# Learning to Reason for Long-Form Story Generation

**Alexander Gurung, Mirella Lapata**
School of Informatics
University of Edinburgh
Edinburgh, UK
a.gurung-1@sms.ed.ac.uk, mlap@inf.ed.ac.uk

## Abstract

Generating high-quality stories spanning thousands of tokens requires competency across a variety of skills, from tracking plot and character arcs to keeping a consistent and engaging style. Due to the difficulty of sourcing labeled datasets and precise quality measurements, most work using large language models (LLMs) for long-form story generation resorts to combinations of hand-designed prompting techniques to elicit author-like behavior. This is a manual process that is highly dependent on the specific story-generation task. Motivated by the recent success of applying RL with Verifiable Rewards to domains like math and coding, we propose a general story-generation task (Next-Chapter Prediction) and a reward formulation (Verifiable Rewards via Completion Likelihood Improvement) that allows us to use an unlabeled book dataset as a learning signal for reasoning. We learn to reason over a story's condensed information and generate a detailed plan for the next chapter. Our reasoning is evaluated via the chapters it helps a story generator create, and compared against non-trained and supervised fine-tuning (SFT) baselines. Pairwise human judgments reveal the chapters our learned reasoning produces are preferred across almost all metrics, and the effect is more pronounced in Sci-Fi and Fantasy genres.[1]

## 1 Introduction

Long-form story generation is a difficult modeling task that requires synthesizing thousands of tokens of rich and subtle text into coherent and character-aware narratives. High-quality writing balances interesting characters with engaging world-building and satisfying plot arcs (Jarvis, 2014; Kyle, 2016), posing problems of long-term dependencies and accurate Theory-of-Mind modeling. Although Large Language Models (LLMs) have shown recent promise in writing long texts (Yang et al., 2022; 2023; Bai et al., 2025; Xie et al., 2023; Shao et al., 2024a), their stories still struggle on a variety of criteria like originality, plot, character development, and pacing (Chakrabarty et al., 2024; Wang et al., 2023; 2022; Ismayilzada et al., 2024), in addition to more fundamental long-form generation flaws like repetition and quality degradation (Que et al., 2024; Wu et al., 2024).

Outside of long-form story generation, Reinforcement Learning (RL) has been successfully applied to post-training LLMs across various tasks, from optimizing general human preferences to improving performance in specialized domains like code generation and math (Li et al., 2025; Lambert et al., 2024; Kumar et al., 2025). Although some RL work briefly touches on creative writing, these efforts typically focus on short-form tasks and are framed as an overall preference alignment (Nguyen et al., 2024; Zhao et al., 2024). In general, RL methods for LLMs have necessitated at least one of the following: (a) *high quality datasets* labeled with preferences, rewards, or verified reasoning traces or (b) *high-quality reward models*, often in the form of verifiable reward functions. Large-scale datasets have yielded strong results for broader tasks like human-preference-alignment (Rafailov et al., 2023; Ethayarajh et al., 2024;

---

[1]We release reproduction and training code at github.com/Alex-Gurung/ReasoningNCP.

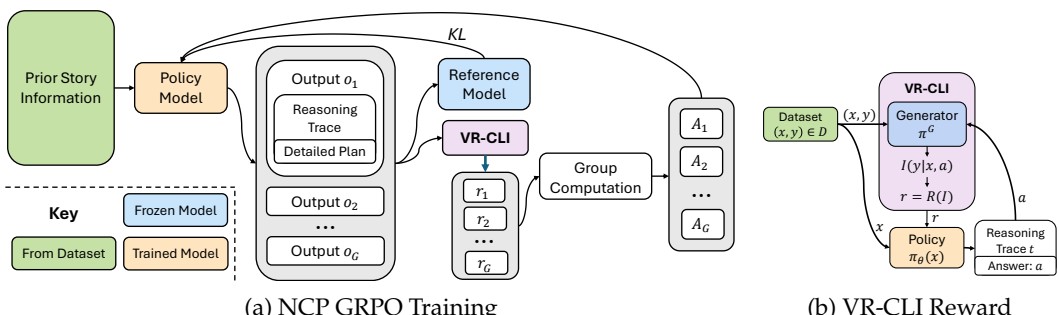

Figure 1: Next-Chapter Prediction (NCP) training procedure (a) using our VR-CLI reward paradigm (b), with GRPO (Shao et al., 2024b). Our reward uses a reference model to get the improved likelihood of the true next chapter. The reference model is a copy of the policy model frozen at the start of training, used both as our story generator $\pi^{\mathcal{G}}$ and for computing KL-divergence. The policy model is our reasoning model $\pi_\theta^{\mathcal{R}}$, trained to produce detailed plans of the next chapter. VR-CLI is described in Section 6 and training in Section 5.

Cui et al., 2024; Wang et al., 2025a), while recent efforts in the math and coding domains have achieved significant success through the use of high-quality rewards (Gehring et al., 2025; DeepSeek-AI et al., 2025; Shao et al., 2024b; Lambert et al., 2024; Kimi Team et al., 2025; Li et al., 2025; Kumar et al., 2025). In particular, there has been a recent surge of interest in enhancing the reasoning capabilities of LLMs using verifiable rewards (Lambert et al., 2024). Instead of relying on a learned reward model, recent work uses deterministic verification functions — usually rule-based methods that assess answers for correctness and formatting.

Neither approach is practical for long-form story generation: story 'correctness' is ill-defined, and collecting large datasets of labeled story completions is challenging. The multifaceted and nuanced nature of narratives also makes the formulation of singular reward functions difficult. Instead, the 'quality' of stories or their continuation is usually assessed through pairwise comparisons across multiple metrics (Yang et al., 2022; Huot et al., 2025; Yang et al., 2023; Xie & Riedl, 2024). Most current work on story generation has therefore avoided RL, and instead leaned into hand-designed systems that mimic different parts of the human writing process, like drafting, editing, and planning. (Fan et al., 2018; Zhou et al., 2023; Wang et al., 2023; Xie & Riedl, 2024; Wang et al., 2025b). Huot et al. (2025) train LLMs to model human story preferences across metrics like creativity and language use, but they do not train story generation models using these evaluators.

Rather than predicting and evaluating entire book-length generations, we propose **Next-Chapter Prediction** (NCP) as a more tractable and informative task. Inspired by the human book-writing process (Kyle, 2016; Jarvis, 2014) that uses both a high-level sketch of the story and more fine-grained information about characters and plot progression, we model story generation as predicting the next chapter given a similar collection of story information. Fine-tuning directly on this task, quickly leads to overfitting and fails to capture the underlying reasoning behind the story-writing process. Instead, we propose a novel reasoning paradigm akin to Reinforcement Learning from Verifiable Rewards (RLVR; Lambert et al. 2024) that allows RL training of reasoning traces for story generation.

We introduce **Verifiable Rewards via Completion Likelihood Improvement** (VR-CLI), a reward modeling paradigm designed to learn reasoning traces that enhance a generator's ability to reproduce a given dataset (see Figure 1). Our key assumption is that increasing the generator's likelihood of predicting the next chapter will, in turn, improve the quality of its generations. Accordingly, our reward is defined with respect to the improvement in predicting a gold next chapter, which can be naturally expressed in terms of per-token perplexity. We use VR-CLI to train Qwen 2.5 models (Qwen et al., 2024) to produce traces that cite and reason about the given story information before constructing a detailed plan, which is then fed as context to the base model for generating the chapter continuation. Our

results show that these models outperform reasoning and non-reasoning baselines and models trained via SFT on next-chapter prediction. Our contributions are as follows:

- We introduce Next-Chapter Prediction, a new task for long-form creative writing.
- We propose VR-CLI, a proxy reward formulation for reasoning that relies only on a high-quality dataset to mimic completions.
- We show that training using this objective improves the generated next-chapters, as judged in pairwise human evaluations.

## 2 Related Work

**Story Generation** Story generation has long been studied as a completion task (Mostafazadeh et al., 2016; Fan et al., 2018), but only with recent advances in long-context modeling have language models begun producing stories of considerable length. A large body of previous work has shown that incorporating condensed story information, such as plot outlines (Fan et al., 2018; Zhou et al., 2023; Wang et al., 2023; Xie & Riedl, 2024; Wen et al., 2023; Yoo & Cheong, 2024), as well as setting and character details (Yang et al., 2022; 2023), significantly improves generation quality. Further improvements have been achieved by explicitly breaking down the writing process into sub-tasks handled by different 'agents' or models (Huot et al., 2025; Peng et al., 2022). However, these approaches largely rely on hand-crafted prompting techniques to generate and refine both the condensed story information and the story itself. Progress in long-form story generation has been hindered by challenges in evaluation, which is inherently subjective, nuanced, and time-consuming. Human evaluation has become the de facto standard for assessing machine-generated stories, typically through pairwise judgments across key dimensions such as coherence, plot development, creativity, and characterization (Yang et al., 2022; 2023; Huot et al., 2025; Xie & Riedl, 2024; Chhun et al., 2022).

**Reinforcement Learning for LLM Reasoning** Fine-tuning LLMs with Reinforcement Learning (RL) has become an increasingly popular method for improving performance on a variety of tasks. Traditionally, RL has been used to align model generations with human preferences (RLHF; Grattafiori et al. 2024; Achiam et al. 2023; Ziegler et al. 2019; Ouyang et al. 2022; Stiennon et al. 2020; Christiano et al. 2017). Building on these online methods, recent work has introduced algorithms that can also learn policies from static preference datasets (Rafailov et al., 2023; Ethayarajh et al., 2024).

RL techniques have further led to significant advancements in verifiable and executable domains such as math, science, and coding (Gehring et al., 2025; Simonds & Yoshiyama, 2025; Li et al., 2025; Kumar et al., 2025). Several recent large-model releases have also emphasized the use of RL to improve performance on math and programming benchmarks (DeepSeek-AI et al., 2025; Kimi Team et al., 2025). Many of these approaches build upon the Reinforcement Learning from Verifiable Rewards (RLVR) paradigm introduced in Lambert et al. (2024), which trains LLMs to produce useful reasoning traces for tasks with easily verified answers via a binary or scaled reward.

We are not aware of previous work that learns reasoning traces for creative long-form generation, possibly due to the challenge of defining rewards with objective correctness criteria. However most similar in approach, Hu et al. (2024a) use LLM-based GFlowNets (Bengio et al., 2021) for sentence continuation and 5-sentence-story infilling using the ROCStories dataset (Mostafazadeh et al., 2016). By representing many LLM tasks as latent variable modeling problems, they connect RL approaches to the field of probabilistic inference.

Concurrent work has also begun exploring the idea of using a model's likelihood of the solution to optimize reasoning. JEPO (Tang et al., 2025), VeriFree (Zhou et al., 2025), LATRO (Chen et al., 2024) and INTUITOR (Zhao et al., 2025) all propose similar policy optimizers based on this latent-variable perspective, and largely apply their methods to math tasks. Our VR-CLI formulation differs in its reward shaping, lack of SFT-objective, and its application to long-form creative writing.

| Story Information | Sample Sentence |
|---|---|
| Global Story Sketch | As the story progresses, it becomes clear that Barbara's family is complex and troubled. |
| Previous Story Summary | As the campers settle into their routine, Tracy becomes friends with Barbara, who arrives at the camp on foot (Chapter 7). |
| Character Sheets | She wants to be a good mother to Bear and seeks validation through her role as a mother. |
| Previous Chapter | The implication, Alice understood, was that to have more than one boy would complicate matters when it came to passing on the bank. |
| Next Chapter Synopsis | Peter tells Carl that his son is missing. |
| Next Chapter | And when a voice came through the wires, it wasn't a member of the staff, but Peter Van Laar himself—to whom Carl nodded each time they crossed paths at the Preserve, but to whom he had actually spoken maybe twice in his life. |

Table 1: Sample sentences from each Story-Information element; data for *The God of the Woods* (by Liz Moore), Chapter 15. Note the different style of text (e.g., detailed prose vs high-level descriptions) and different focus (e.g., character-focused vs plot-focused).

## 3  Next-Chapter Prediction

Instead of generating and evaluating entire book-length stories at once, we propose Next-Chapter Prediction (NCP) as a more tractable task. We draw inspiration from the methods real authors adopt when writing a book (Kyle, 2016; Jarvis, 2014) which may involve creating a high-level global story outline, along with more detailed plans for individual chapters.

We define the 'writing process' as following this general structure: first, an author creates a high-level sketch of the story. As the writing progresses the author refines their plot and character development. Before writing each chapter, the author prepares a short synopsis of what should occur. Based on this writing process, we formalize our NCP task as follows.

Let $SI_i$ collectively denote *Story-Information* at chapter index $i$. Specifically, $SI_i$ represents:

- A global sketch of the entire story (see Table 1);
- A summary of the previously written chapters (see Table 1);
- Character sheets, based on previously written chapters (see Table 1);
- Previous story text, which we approximate by the previous chapter (see Table 1).
- Synopsis of the next chapter (see Table 1);

At a chapter index $i$ with Story-Information $SI_i$, we denote the predicted next chapter as $\hat{c}_{i+1}$. When the next chapter is given, not predicted, we denote it $c_{i+1}$. We denote the *story-generator model* that predicts the next chapter $\pi_\theta^{\mathcal{G}}$. The default setup for this task is to directly generate the next chapter from the story information, using the story generator:

$$\hat{c}_{i+1} \leftarrow \pi_\theta^{\mathcal{G}}(SI_i) \tag{1}$$

Our claim in this work is that reasoning traces can lead to *better* next-chapter prediction; specifically, we use a *reasoning model* $\pi_\theta^{\mathcal{R}}$ to predict a reasoning trace about the given story information, ending with a more detailed plan for the next chapter. We denote this predicted plan $\hat{p}$. In reasoning model variants, we generate the next chapter using a story-generator model conditioned on both the story information and the plan:

$$\hat{p} \leftarrow \pi_\theta^{\mathcal{R}}(SI_i) \tag{2} \qquad\qquad \hat{c}_{i+1} \leftarrow \pi_\theta^{\mathcal{G}}(SI_i, \hat{p}) \tag{3}$$

We evaluate the effectiveness of story-generation and reasoning models via the quality of the next chapter, judged both on its merit and its ability to fit into the broader story context. Note that our proposed reasoning method optimizes the detailed plans, which are then used to improve the generated chapters. More details are provided in Section 7.2, and example reasoning traces are in Appendix G.

## 4 Dataset Curation

We collect a dataset of 30 books published in or after 2024 (to mitigate information leakage).[2] Details on book selection are in Appendix A.1. Our books range from 67k to 214k tokens, with a mean of 139k. Rather than raw story text we operate on higher-level plot and character representations as prior research has shown these to be useful for story tasks (Pham et al., 2025; Gurung & Lapata, 2024). We follow previous work in sourcing *gold-standard* chapter summaries from SuperSummary[3] (Xu et al., 2024; Yuan et al., 2024; Ou & Lapata, 2025). We explain below how we use these in building the story-information $SI_i$.

### 4.1 Story Information

We denote the following story information $SI_i$, defined at a given story and chapter index $i$.

**High-level Story Sketch** This can be thought of as an author's sketch of how the story should unfold. As we do not have these sketches available, we mimic them by summarizing all of a book's individual chapter summaries together into a high-level plan (see Table 1). We use Llama 3.3 70B (Grattafiori et al., 2024) to generate these summaries.

**Previous Story Summary** At a given chapter index $i$, we also summarize the previous chapter summaries $\leq i$ to give a more detailed representation of what has already been written (see Table 1). Again we use Llama 3.3 70B. Hyperparameter details are in Appendix A.2.

**Character Sheets** Prior work has shown that that character information is useful for downstream story tasks (Gurung & Lapata, 2024; Huot et al., 2025), so we incorporate character information into $SI_i$ by adapting the CHIRON character sheets for longer stories (Gurung & Lapata, 2024). These sheets contain four broad categories of character information and are automatically generated from story text before being filtered for accuracy using an entailment module. We create individual character sheets for the three main characters of each story, defined at each chapter index $i$ based on the previous chapters $\leq i$. We use the Llama 3.3 70B model for both generation and entailment modules and add a summarization step to consolidate the sheets. More details are in Appendix A.3.

**Next Chapter Synopsis** For a given chapter index $i$, we call the summary of the next chapter a 'synopsis'. This contains a rough idea of what should happen in the chapter, which informs the path the story generations take. On average, the synopsis is 7.4% of the size of the next chapter (in tokens). We hypothesize that by adding detail to the synopsis via reasoning (see Table 1), we can more effectively guide the generation of the chapter.

### 4.2 Dataset Statistics

The final dataset has 30 books, split 22-4-4 into training, validation, and testing. We split by book to evaluate our method's generalization capabilities across books and authors, as training may learn book or author-specific style or plot information. Further splitting into chapters and lightly filtering for chapter length ($200 \leq \#$ words $\leq 5,000$) and chapter location ($2 < i < |S| - 2$) gives us 1,004 training datapoints, 162 for validation, and 181 for testing. More details are provided in Appendix A.4 and Tables 6, and 7. Example sentences from each element in our dataset are in Table 1.

## 5 Verifiable Rewards via Completion Likelihood Improvement (VR-CLI)

As explained earlier, it is challenging to define a reward model for long-form creative writing in part due to the length of generations and the subtle nature of the task. Furthermore, collecting labeled datasets of story continuations is costly due to the sheer variety of possible

---

[2]Our work primarily uses two models: Llama 3.3 70B and Qwen 2.5 3B/7B-1M Instruct. The Llama model has a knowledge cutoff of December 2023, and while the Qwen models were released in 2024, we find little evidence of data contamination.

[3]https://supersummary.com/

story continuations. We propose circumventing these constraints by creating a proxy reward that incentivizes useful reasoning steps, utilizing the book dataset described in Section 4.

Taking inspiration from RLVR (Lambert et al., 2024) and the latent-variable modeling framing of Hu et al. (2024a), we introduce **Verifiable Rewards via Completion Likelihood Improvement (VR-CLI)**. VR-CLI is a reward modeling paradigm that learns reasoning traces that help a generator reproduce a given dataset without the need for labeled data or defined external rewards. This formulation relies on a key assumption: improving the generator's likelihood of producing the provided dataset will, in turn, improve the quality of its generations. We verify this assumption for NCP in Section 8. We believe VR-CLI is a promising reward paradigm for other tasks where completions are not easily verified.

Let $D$ represent a dataset of pairs $(x, y)$, where $x$ is the input prompt, and $y$ is the gold completion. A reasoning model generates a reasoning trace $t$ which culminates in a final answer $a$. We define a generator model, $\pi^{\mathcal{G}}$, that is not trained but instead used to get the likelihood of generating $y$. For our NCP task, $x$ is the story information, $y$ is the gold next chapter, $a$ is the detailed plan $\hat{p}$, and our generator $\pi^{\mathcal{G}}$ is the story generator model.

We define the improvement $I$ as the *percent improvement in per-token perplexity (PPL)* when generating $y$ from $\pi^{\mathcal{G}}$, conditioned on $(x, a)$, compared to the perplexity of $y$ conditioned only on $x$. This measure indicates how much the inclusion of this answer increases the likelihood of generating $y$. Note that the perplexity is calculated from the probability distribution over tokens in $y$, not the probability of the entire text. Additionally, the prompt formulation can differ between $y|x$ and $y|x, a$ as necessary for the specific subtask.

$$I_{\pi^{\mathcal{G}}}(x, y, a) = \left[\frac{PPL_{\pi^{\mathcal{G}}}(y|x) - PPL_{\pi^{\mathcal{G}}}(y|x, a)}{PPL_{\pi^{\mathcal{G}}}(y|x)}\right] \times 100 = \left[1 - \frac{PPL_{\pi^{\mathcal{G}}}(y|x, a)}{PPL_{\pi^{\mathcal{G}}}(y|x)}\right] \times 100 \quad (4)$$

Positive values imply the perplexity *with reasoning* is lower than the perplexity without, and the reasoning improved the likelihood of generating the gold continuation. Negative values imply the opposite, that the reasoning worsened the likelihood. Note that $PPL_{\pi^{\mathcal{G}}}(y|x)$ does not depend on the policy model or answers, so it can be pre-computed before training. Although we use perplexity as our likelihood measure in this work, a very similar formulation could be constructed with average log-likelihood instead. We found both methods to produce similar results; more details are presented in Appendix A.8.

There are many ways to define our reward based on the improvement $I$, for example: (1) as a piecewise function, like RLVR, subject to the magnitude of the improvement and custom thresholds ($\{\omega_0, \omega_1, ... \omega_n\}$) or (2) as a raw value bounded by 0 when necessary.

$$R(x, y, a) = \begin{cases} \alpha & I(x, y, a) \leq \omega_0 \\ \beta & \omega_0 < I(x, y, a) \leq \omega_1 \quad (5) \\ \gamma & \omega_1 < I(x, y, a) \end{cases} \qquad R(x, y, a) = \max[0, I(x, y, a)] \qquad (6)$$

The specific reward formulation will depend on both the downstream task and the RL-training algorithm used. Figure 1b illustrates the high-level reward paradigm. The reward and training algorithm we use for our NCP task are described in the following sections.

Additionally, one could forgo the baseline and use the chosen likelihood-measure directly (e.g. $I_{\pi^{\mathcal{G}}}(x, y, a) = -PPL(y|x, a)$). Notably, when using GRPO's advantage estimation and a simple reward equal to improvement ($R(x, y, a) = I(x, y, a)$), advantages are equivalent with and without the baseline (Shao et al., 2024b). However, we hypothesize that including a baseline provides a more interpretable and useful reward estimate, especially when using a more complicated reward definition. For example, across a group where all samples produce worse-than-baseline perplexities, it may be useful to set their rewards uniformly to zero (and not increase the likelihood of potentially damaging traces). We also test this in ablations in Appendix A.8 and found comparable results, and believe both formulations are worth exploring in future tasks. As all of these formulations retain the goal of increasing the gold-completion's likelihood via the reward, we view them as variants of the VR-CLI paradigm.

# 6 Reinforcement Learning for Next-Chapter Prediction

**Reward Modeling for Next-Chapter Prediction**  We use the thresholded VR-CLI reward paradigm introduced in Equation (5) to define our reward for each (story-information, chapter, reasoning) tuple $(SI_i, c_{i+1}, \hat{p})$. Note that our 'answer' is the detailed plan $\hat{p}$ our policy model produces after reasoning over the story information. Example reasoning traces are in Appendix G. We use the reference model for our policy as our story-generator, $\pi^{\mathcal{G}}$, as we assume that the skills needed for story-reasoning and summarizing are different than those needed for engaging story-writing. For other tasks where the reasoning and answering skills are not so distinct it may make sense to use the policy model as the generator as well.

Applying VR-CLI to our task implies that plans that induce a higher likelihood of the true next chapter will also produce better generated chapters. We validate this assumption in Section 8, but also believe it aligns with our intuition concerning useful plans. For example, plans that correctly predict plot events, character details, or writing style in the true next chapter should increase its likelihood, and should also be a useful basis for generating a novel chapter. We provide example aligned excerpts between plans and next-chapters in Table 19.

We define our thresholded reward as:

$$R(SI_i, c_{i+1}, \hat{p}) = \begin{cases} 0, & I(SI_i, c_{i+1}, \hat{p}) < 0.05 \\ 0.5, & 0.05 \le I(SI_i, c_{i+1}, \hat{p}) < 1 \\ 0.9, & \text{if } 1 \le I(SI_i, c_{i+1}, \hat{p}) < 2 \\ 1, & \text{if } I(SI_i, c_{i+1}, \hat{p}) \ge 2 \end{cases} \tag{7}$$

We find that this mild reward shaping helps ensure consistent training trajectories by encouraging both 1) large perplexity improvements and 2) promising samples at the early stages of training when significant improvements in perplexity are rare. As we load the reference model for calculating the KL-divergence, our reward requires minimal computational overhead. Future work could explore lightweight reasoning for strong story generators (or vice-versa), but our current setup significantly reduces training complexity.

**GRPO Training**  Our task and reward setup are relatively RL-algorithm agnostic; one could use an offline reasoning-reward dataset or any online policy learning algorithm that works with a verifiable reward. We hypothesize that online methods are preferable for NCP as reasoning styles change over training, and preliminary tests show low reward prior to training: initially, average improvement is -0.06% with Qwen 2.5 7B.

We use GRPO for RL training as it has seen recent success in verifiable reward domains (Shao et al., 2024b; DeepSeek-AI et al., 2025) and reduces computational overhead by removing the value model. GRPO generates multiple responses for each prompt and normalizes each reward by its group, before calculating the advantages and updating the policy parameters $\theta$. See Figure 1a for a training overview and Appendix A.6 for more details.

# 7 Experimental Setup

## 7.1 Model Comparisons

Prior work has found success using the Qwen model series (Qwen et al., 2024), and Gandhi et al. (2025) showed they exhibit cognitive behaviours useful for reasoning. The majority of our experiments use Qwen-2.5 7B-Instruct-1M as our base model for training and story generation. We also report results with 3B, to examine trends across model sizes. Future work could further scale these experiments, but due to the long context size, large-model training is computationally expensive. For clarity, models $\pi_\theta$ with noted parameters $\theta$ are trained, and models $\pi$ without noted parameters are frozen without further training. More training details are in Appendix A.6. We compare the following model variants.

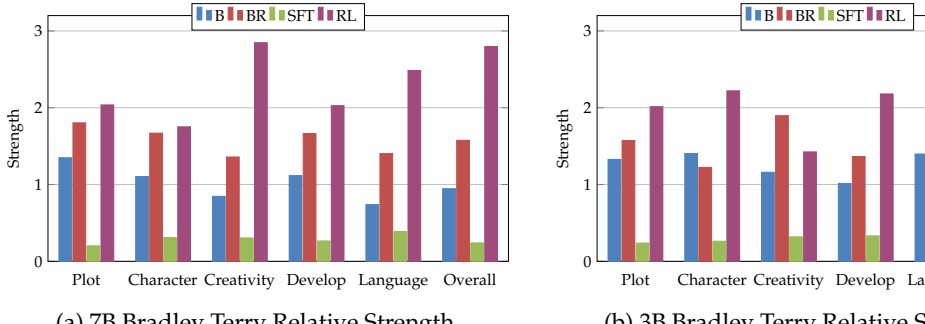

(a) 7B Bradley Terry Relative Strength    (b) 3B Bradley Terry Relative Strength

Figure 2: Bradley Terry relative strength parameters for each variant, by model size. RL-Trained (RL) has the highest relative strength in most dimensions, and the effect is more pronounced in 7B models. Table 2 shows win-probabilities for 7B models. Variant shorthands are introduced in Section 7.

**RL-Trained (RL)** Our proposed method, training a policy model with VR-CLI and GRPO. Our story-generator is not further trained, but our policy model is trained to produce useful plans via our VR-CLI reward (Section 6): $\hat{p} \leftarrow \pi_\theta^\mathcal{R}(SI_i); \ \hat{c}_i \leftarrow \pi^\mathcal{G}(SI_i, \hat{p})$.

**Base (B)** We compare RL-Trained against a baseline model which does not train our story-generator or produce any reasoning. This is our default NCP variant: $\hat{c}_i \leftarrow \pi^\mathcal{G}(SI_i)$.

**Base-Reasoning (BR)** We also compare to a model which attempts to predict the next chapter by first generating a reasoning trace; we do not train our story-generator or our reasoning model (they are both set to the same): $\hat{p} \leftarrow \pi^\mathcal{R}(SI_i); \ \hat{c}_i \leftarrow \pi^\mathcal{G}(SI_i, \hat{p})$.

**Supervised Fine-Tuning (SFT)** We fine-tune our story-generator on the NCP task with the SFT objective on next chapters. The generator predicts chapters like Base: $\hat{c}_i \leftarrow \pi_\theta^\mathcal{G}(SI_i)$.

## 7.2 Human Evaluation

We evaluate the generated story continuations by eliciting pairwise preferences following evidence suggesting that relative judgments can be more reliable than absolute ones (Louviere et al., 2015; Stewart et al., 2005; Liu et al., 2024). We draw on the criteria outlined in previous work (Huot et al., 2025; Chakrabarty et al., 2024; Chhun et al., 2022) to evaluate the continuations along the following dimensions: (1) **Plot:** Does it exhibit events and turns that move the plot forward logically? (2) **Creativity:** Does the continuation have engaging characters, themes, and imagery, and avoid overly cliched characters and storylines? (3) **Development:** Does it introduce characters and settings with appropriate levels of detail and complexity? (4) **Language Use:** Is the language varied and rich, exhibiting rhetorical, linguistic, and literary devices? (5) **Characters:** Does it feature believable and conceptually consistent characters, including reasonable character arcs and development? (6) **Overall Preference:** Which of the two continuations did you prefer? Full instructions and recruitment details are in Appendix C. Annotators were recruited through Prolific if their primary language was English and their employment role was in creative writing.

To validate our annotation procedure, we compute Fleiss' kappa across a dataset of Base vs. Base-Reasoning comparisons. We find fair agreement across annotation dimensions, with the highest agreement in Creativity and Overall Quality. More details are in Appendix D. Initial experiments showed gold-continuations were almost exclusively preferred to model generations, so we exclusively collect inter-variant comparisons.

We randomly sample 5 chapter indices per story in our test set and generate continuations for each model. Participants are shown the story information and two possible continuations and asked to provide preferences along the above dimensions. We collect one judgment per datapoint (20 datapoints per variant-comparison), and convert the preferences into relative model strengths using a Bradley-Terry model (Bradley & Terry, 1952).

| Dimension | SFT-B | BR-B | BR-SFT | RL-SFT | RL-B | RL-BR |
|---|---|---|---|---|---|---|
| Plot | 5.3 | 72.2 | 83.3 | 89.5 | 52.9 | 62.5 |
| Character | 26.3 | 68.8 | 84.2 | 89.5 | 46.7 | 61.5 |
| Creativity | 21.1 | 57.1 | 82.4 | 84.2 | 81.2 | 70.6 |
| Development | 22.2 | 68.8 | 88.2 | 89.5 | 52.9 | 66.7 |
| Language | 35.0 | 66.7 | 76.5 | 88.9 | 80.0 | 58.8 |
| Overall | 15.0 | 66.7 | 83.3 | 90.0 | 76.5 | 64.7 |

Table 2: Bradley-Terry preference probabilities (%) that A is preferred over B in A-B pairwise comparisons. RL-Trained is preferred across almost all dimensions, and against Base has an overall 76.5% preference probability. See Section 7 for variant details.

| Genre | BR | RL | Diff |
|---|---|---|---|
| Sci-Fi | **0.68** | **1.68** | **1.00** |
| Fantasy | 0.50 | 1.11 | 0.61 |
| Romance | 0.27 | 0.91 | 0.64 |
| Historical | -0.33 | 0.60 | 0.93 |

Table 3: Mean percent improvement by genre on our test set. 'Diff' is the additional percent improvement gained by RL-training our reasoning. Model shorthands are in Section 7.

| Model | # Words | Unique Words | Unseen Trigrams | Rouge-L F1 | Rouge-L Prec |
|---|---|---|---|---|---|
| Base | 1486.5 | 32.7% | 77.8% | 0.10 | 0.286 |
| SFT | 1278.1 | 23.7% | 53.8% | 0.09 | 0.308 |
| Base-Reasoning | 1568.0 | 31.8% | 77.5% | 0.11 | 0.282 |
| RL-Trained | 1479.5 | 33.2% | 77.2% | 0.11 | 0.306 |

Table 4: We calculate automated measures of length and diversity from the generated chapters across 7B model outputs: # Words (average number of words per chapter), % Unique words within a chapter, % Unseen Trigrams (% of all trigrams in a chapter not present in *S1*). Rouge-L (F1 and Precision) is computed against the gold standard, reference chapters. We find broadly similar results and uniqueness (Appendix B), indicating that the preferences come from higher quality storytelling instead of longer stories.

## 8 Results

We report both Bradley-Terry based probabilities (Table 2) and relative strengths (Figure 2), as well as true win-rates, including ties (see Table 15). We also investigate surface differences in predicted chapters (Table 4) and percent-improvement by genre (Table 3).

**Training reasoning improves overall performance.** We find our RL-Trained 7B model is overwhelmingly preferred over all three baseline variants and has the highest relative strength value for all dimensions. Figure 2 shows the relative strength scores for each comparison model. RL-trained yields the biggest improvements in creativity and overall preference, and the lowest improvement in characters. This model has a 64.7% preference probability over Base-Reasoning in Overall Quality, indicating that optimizing our reward aligns with human judgments (see Table 2).

**SFT model performs poorly on next chapter prediction.** We find significant repetition issues in the chapters generated from the SFT model. For a fair comparison, we automatically truncate clear mode-collapse repetitions (Appendix B), but annotators still strongly prefer all other variants. We hypothesize that SFT performance would be improved by training on a much larger dataset to reduce the effect of overfitting; doing so may also be helpful as a starting point for reasoning models (DeepSeek-AI et al., 2025). We therefore believe that our method is more effective in low-resource settings, but can be used in concert with SFT approaches when more data is available.

**Differences between models go beyond surface-level variations.** We investigate surface-level differences between the generated chapters by comparing their length, word/trigram diversity. While we do find the SFT-based chapters to be significantly shorter and less lexically diverse (Table 4), we find few statistical differences between the other models. This indicates that annotators were not biased towards longer/more varied stories and instead judged based on story content. Table 9 and Appendix B contain more details.

**Perplexity improvement correlates with human judgments.** To evaluate VR-CLI's Improvement's utility as a proxy metric for human judgments, we compute the correlation between the pairwise judgment and the Improvement given a reasoning trace and gold next-chapter (Appendix D.1). We find a significant and positive Spearman's rank correlation of ($\rho = 0.33, p = 0.01, N = 60$). While this improvement metric cannot be computed at true test time (without a gold continuation), it can be a useful proxy for further evaluating reasoning models on an unlabeled dataset without collecting expensive human judgments.

**Reasoning has the most pronounced effect on the Sci-Fi genre.** We compare pairwise preferences and average percent improvement across different genres. We find that reasoning has a particularly strong positive effect on Sci-Fi, and that training further improves the effect. Table 3 shows the percent improvement for Base-Reasoning and RL-Trained variants. See Tables 17 & 18 for pairwise preferences broken down by model size and genre.

**Reasoning improves bigger models more.** To test if these trends are exclusive to 7B models, we also perform the same experiments using Qwen 2.5 3B as a base. Relative strength scores are shown in Figure 2b and preference probabilities in Table 13; our RL-Trained setting still outperforms the baselines across most dimensions, but the effect is smaller and annotators preferred Baseline-Reasoning in 'Creativity' and 'Language' dimensions. We find similar genre effects (Sci-fi still has the highest percent-improvement) and limited lexical differences between settings; more details are in Appendix F. Due to the greater relative strength of RL-Trained 7B variants compared to their non-trained variants, we hypothesize that larger models are better able to learn useful reasoning for this task.

## 9 Conclusion

We approach the challenge of long-story generation by proposing an author-inspired story-generation task, Next-Chapter-Prediction (NCP), that uses a combination of high-level and locally relevant story information to predict the next chapter of a book. We evaluate performance on this task via a dataset of recently published books and their per-chapter summaries, from which we derive a useful dense collection of story information.

We introduce a novel reward modeling framework, Verifiable Rewards via Completion Likelihood Improvement (VR-CLI), that allows us to optimize a reasoning model to predict plans that increase the likelihood of the next chapter. We train 7B and 3B Qwen models using this framework and GRPO, and show that the resulting story completions are strongly preferred by expert judges over baselines. In the future, we plan to apply our VR-CLI framework to generation tasks whose output cannot be easily verified, such as long-form summarization and question answering.

## Acknowledgments

We thank the anonymous reviewers for their constructive feedback. We gratefully acknowledge the support of the UK Engineering and Physical Sciences Research Council (grant EP/W002876/1). We would also like to thank Kolya Malkin for their advice and support on this work.

## Ethics Statement

Story generation systems have the potential to perpetuate biases (e.g., stereotypical characters) and create harmful content. As the training signal for our work comes from already-published books we hope our models do not learn any more harmful behaviour than is already present in their pretraining data. However, future work should explore ways to measure and mitigate the biases of LLMs in long-form story generation.

Research on text-generation systems also needs to be careful to avoid violating the intellectual property rights of the authors whose work is being used. As our dataset (and therefore the models trained on it) are created using copyrighted work we individually purchased,

we are unable to publicly release our dataset or models. We will however release the code necessary to reproduce our work, and the books used are listed in Table 5.

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

## A  Data Curation

### A.1  Book Selection

We selected fiction books based on popularity, critical acclaim, and the availability of chapter summaries on SuperSummary. We excluded all sequels and books based on previous stories (e.g., retellings of myths) to ensure that all relevant information was contained within the book itself. We also excluded collections of short stories, stories featuring existing characters (e.g., a known detective character), and stories relying on visual information (e.g., graphic novels). We also aimed to include a variety of genres; our train/test/validation set all satisfy the following conditions:

1. At least one Sci-Fi book (that is not also fantasy), one fantasy book (that is not also Sci-Fi), and one book that is neither
2. One historical-fiction book
3. One romance book
4. One young-adult book, and one adult book

These conditions are meant to ensure that when we test for generalization we are not overfitting to a specific genre/type of book.

This process gave us the books listed in Table 5.

| Title | Author | Release Date | Split |
|---|---|---|---|
| The Women | Kristen Hannah | February, 2024 | Train |
| Funny Story | Emily Henry | April, 2024 | Train |
| First Lie Wins | Ashley Elson | January, 2024 | Train |
| Just for the Summer | Abby Jimenez | April, 2024 | Train |
| When the Moon Hatched | Sarah A. Parker | May, 2024 | Train |
| The Familiar | Leigh Bardugo | April, 2024 | Train |
| We Solve Murders | Richard Osman | September, 2024 | Train |
| The Book of Doors | Gareth Brown | February, 2024 | Train |
| I Cheerfully Refuse | Leif Enger | April, 2024 | Train |
| You are Here | David Nicholls | April, 2024 | Train |
| The Reappearance of Rachel Price | Holly Jackson | April, 2024 | Train |
| Martyr | Kaveh Akbar | January, 2024 | Train |
| Deep End | Ali Hazelwood | February, 2024 | Train |
| The Husbands | Holly Gramazio | April, 2024 | Train |
| Blue Sisters | Coco Mellors | May, 2024 | Train |
| Sandwich | Catherine Newman | June, 2024 | Train |
| A Fate Inked in Blood | Danielle L. Jenson | Febuary, 2024 | Train |
| Heartless Hunter | Kristen Ciccarelli | February, 2024 | Train |
| Water Moon | Samantha Sotto Yambao | January, 2024 | Train |
| Beautiful Ugly | Alice Feeney | January, 2024 | Train |
| The Three Lives of Cate Kay | Kate Fagan | January, 2024 | Train |
| All the Colours of the Dark | Chris Whitaker | June, 2024 | Train |
| Burn | Peter Heller | August, 2024 | Val |
| The Warm Hands of Ghosts | Katherine Arden | February, 2024 | Val |
| The Tainted Cup | Robert Jackson Bennett | February, 2024 | Val |
| The Grandest Game | Jennifer Lynn Barnes | July, 2024 | Val |
| The God of the Woods | Liz Moore | July, 2024 | Test |
| The Mercy of Gods | James S. A. Corey | August, 2024 | Test |
| Where the Dark Stands Still | A.B. Poranek | February, 2024 | Test |
| Witchcraft for Wayward Girls | Grady Hendrix | January, 2024 | Test |

Table 5: Books compiled in our dataset, with author names, release dates and partition. We separate books into splits to better test generalization.

## A.2 Chapter Summarization

We generate our high-level story plans and prior-story summaries using the following hyper-parameters (mostly taken from the default generation config):

temperature= 0.6, top_p= 0.9, top_k= 50, max_tokens= $min(4096, |\text{text-tokens}| \cdot 0.8)$

## A.3 Character Sheets

We selected the top three characters based on SuperSummary's Character Analysis details, aiming to represent the main protagonists and antogonists where applicable. When two characters seemed equally important, we broke ties via name occurrences in the corresponding book.

We use the same generation hyper-parameters as the original CHIRON work (Gurung & Lapata, 2024), but adapted to the generation config of Llama 3.3 70B (temperature= 0.6, top_p= 0.9).

We also use the zero-shot entailment module setting with Llama 3.3 70B, relying on its implicit reasoning abilities for the entailment task instead of explicit CoT and ICL prompting. These changes save significant computational resources by removing the need to generate reasoning steps for each claim across every chapter.

| Split | # Datapoints | Avg. N.C. | Max. # N.C. | Avg. Prompt | Max. Prompt |
|-------|-------------|-----------|-------------|-------------|-------------|
| Train | 1004 | 2453.1 | 6677 | 6404.9 | 11216 |
| Val | 162 | 2263.7 | 6732 | 6382.1 | 10433 |
| Test | 181 | 2962.7 | 6217 | 7018.2 | 11176 |

Table 6: Number of datapoints in our final dataset, and corresponding mean/maximum tokens for 1) the next chapter (N.C.) and 2) the prompt to generate reasoning (Prompt) based on the Qwen-2.5 3B tokenizer.

| Story Information | Mean | STD | Min | Max |
|-------------------|------|-----|-----|-----|
| Next Chapter | 2498.8 | 1458.9 | 265 | 6732 |
| Previous Chapter | 2465.7 | 1482.3 | 58 | 6732 |
| Character Sheets | 2072.6 | 234.3 | 776 | 2696 |
| Global Story Sketch | 787.0 | 112.8 | 615 | 1102 |
| Prior Story Summaries | 676.2 | 141.5 | 168 | 1114 |
| Next Chapter Synopsis | 154.21 | 89.12 | 14 | 665 |

Table 7: Size statistics for each element of our Story Information, reported in tokens by the Qwen-2.5 3B tokenizer.

For summarizing the character sheets we use the same hyper-parameters as used for chapter summarization, but with fewer generated tokens: max_tokens= $min(2048, |\text{text-tokens}| \cdot 0.8)$. We summarize each character sheet individually, and concatenate them together in our final story-information prompt. This summarization step drastically reduces the context size needed and makes the character information much denser.

## A.4 Filtering

We filter out datapoints based on the following criteria to reduce training complexity and to ensure our datapoints are meaningful and informative:

- Chapters ($c_{i+1}$) must have $\geq$ 200 words and $\leq$ 5000 words. We also apply the upper limit filter to the previous chapter to reduce the potential size of the prompt.

- We filter out the first two and last two chapters, as they often heavily feature prologue/epilogue text, and are difficult to apply reasoning to (e.g., too little information or too few possibilities). For books at the beginning of a series, the end is also often used to set up future installments.

## A.5 Final Dataset Size

These steps combined give us a final dataset whose descriptive statistics are given in Table 6 and Table 7.

## A.6 Training

Table 8 lists the hyper-parameters for GRPO (top) and SFT (bottom):

More details for reproducing both setups are available in our code. We base our GRPO-training on (Hu et al., 2024b). In both cases we selected the best performing checkpoint via validation performance (percent improvement for GPRO and loss for SFT).

## A.7 SFT Sweeps

We ran small sweeps to test different SFT hyper-parameters, selected from previous work comparing SFT to RL methods. We evaluated learning rates of {2e-5, 5e-7, 5e-6, 4e-5}, and amongst largely similar results found 2e-5 to perform the best by validation loss. Future

| GRPO Hyperparameter | Value |
|---|---|
| Learning Rate | 5e-7 |
| KL Coefficient | 1e-6 |
| Max-Generation-Length | 2048 |
| # Samples per prompt | 16 |
| Rollout batch size | 64 |
| Train batch size | 64 |
| Epochs | 20 |

| SFT Hyperparameter | Value |
|---|---|
| Learning Rate | 2.0e-5 |
| Gradient Accumulation Steps | 64 |
| Epochs | 10 |

Table 8: Hyper-parameters and their values. After training, models were selected by the best-performing checkpoint on validation loss/reward.

work could likely optimize further, but we believe that the largest barrier to high-quality SFT performance is dataset size.

### A.8 Reward Formulation Ablations (VR-CLI Variants)

Recall that for this work, we defined our perplexity-based *Improvement* measure as:

$$I_{\pi^{\mathcal{G}}}(x, y, a) = \left[\frac{PPL_{\pi^{\mathcal{G}}}(y|x) - PPL_{\pi^{\mathcal{G}}}(y|x,a)}{PPL_{\pi^{\mathcal{G}}}(y|x)}\right] \times 100 = \left[1 - \frac{PPL_{\pi^{\mathcal{G}}}(y|x,a)}{PPL_{\pi^{\mathcal{G}}}(y|x)}\right] \times 100$$

and our reward as a piecewise function (although we propose other reward formulations):

$$R(SI_i, c_{i+1}, \hat{p}) = \begin{cases} 0, & I(SI_i, c_{i+1}, \hat{p}) < 0.05 \\ 0.5, & 0.05 \leq I(SI_i, c_{i+1}, \hat{p}) < 1 \\ 0.9, & \text{if } 1 \leq I(SI_i, c_{i+1}, \hat{p}) < 2 \\ 1, & \text{if } I(SI_i, c_{i+1}, \hat{p}) \geq 2 \end{cases}$$

We also ran ablations to test different formulations of our reward, within the same VR-CLI paradigm. Instead of costly human annotations, we evaluate these different methods via their average percent (perplexity) improvement on our test set, using reasoning produced after RL-training. We sample 5 plans to reduce variance. Our "RL-Trained" method, with the previously described improvement and reward formulation, produces an **average percent-improvement of** 0.884.

**NLL for PPL** Our first ablation keeps everything the same, but switches out perplexity for log-likelihood in the *Improvement* calculation. We find a new percent-improvement of 0.485%, about 45% worse than our method. We noticed a lack of convergence with the same hyper-parameters, so we re-ran training for 30 epochs instead of 20, resulting in a percent improvement of 0.627%, still 30% worse.

**Unbounded NLL Improvement** Our second ablation tests a much simpler version of our reward calculation: the *Improvement* calculation is reduced to the log-likelihood (without a baseline calculation, and the 'unbounded' reward is set equal to the improvement.

$$I_{\pi^{\mathcal{G}}}(x, y, a) = \log P_{\pi^{\mathcal{G}}}(y|x,a)$$
$$R(SI_i, c_{i+1}, \hat{p}) = I(SI_i, c_{i+1}, \hat{p})$$

We find slightly better performance with this simpler reward compared to our default setting, with an average percent-improvement of 1.031% (about 16% higher than our method).

This differs from our initial experiments with 3B models (which may have worse reward distributions), which led us to develop the piecewise reward formulation. However, as described in Section 5, one downside to this approach is the worse interpretability during training, as 1) it can be unclear whether performance is better without reasoning, and 2) some samples may dominate the logged metrics.

**Unbounded PPL Improvement** Interestingly, when we performed the same experiment with perplexity ($I_{\pi^{\mathcal{G}}}(x, y, a) = -PPL_{\pi^{\mathcal{G}}}(y|x, a)$) instead of log-likelihood our performance dropped to an average of 0.760.

An interesting avenue for future work is a detailed empirical comparison of these VR-CLI formulations on creative writing tasks and beyond.

### A.9 Lessons from Hyper-parameter Tuning

Due to the high computational cost of our experiments, we were unable to do large hyper-parameter sweeps. We did, however, run many initial tests to gain insight on useful hyper-parameters, which we share here to help guide future research in the space.

We found the following results during our hyper-parameter tuning to be helpful:

**Low KL-divergence coefficient** Similar to recent results in verifiable reward domains (e.g., math; Lambert et al. 2024), we found that a low KL term (the default parameter in some libraries is 0.05; we use 1e-6) improved performance. We hypothesize that lowering this term encourages more exploration by the policy model, and that our reward definition is robust enough that it doesn't require significant regularization. However, we do find some cases of significant repetitions in our reasoning traces, which may be alleviated by 1) training on a larger dataset and 2) increasing the KL-divergence coefficient.

**Longer max-generation lengths** While the majority of our reasoning traces are under 1024 tokens, we found that increasing the maximum generation length to 2048 stopped our reward from converging early to a low value. As we see significant fluctuations in reasoning trace-length during training, we hypothesize that using a higher maximum length prevents potentially informative traces from being cut off and incurring poor reward (as the plan at the end would be malformed).

**Number of samples** We found that increasing the number of generations per sample improved performance even when accounting for the additional computational cost. As described earlier, the initial average improvement is very low, so we hypothesize that increasing the number of generations increases the likelihood of a non-zero reward, allowing the model to learn from more examples at the beginning of training.

In a similar vein, we found it fruitful to spend time tuning the prompt to increase the likelihood of a positive reward from baseline models. We would also encourage future work in data selection during training to select informative data points (Cui et al., 2025; Xie et al., 2025).

## B Chapter Generation

To make model generations more closely comparable to the original chapters and each other, we bound the length of our next-chapter generations to be between half and 1.5 times the length of the original (in tokens): $[0.5 \cdot |c_{i+1}|, 1.5 \cdot |c_{i+1}|]$

We also apply some automatic truncating to the generated chapters to replicate a more realistic scenario. For example, we cut off text after "### End of Chapter" to avoid making judgments on non-next-chapter text. These rules are applied to all completions but induce changes almost exclusively in SFT completions, which often devolve into repeated and unrelated text.

Our automatic formatting rules are:

- Cut off text after end-of-chapter signifiers

| Model | # Words | Unique Words | Unseen Trigrams | Rouge-L F1 | Rouge-L Prec |
|---|---|---|---|---|---|
| 3B-B | 1383.10 | **35.60** | **83.30** | 0.100 | 0.284 |
| 3B-SFT | 1149.30 | 25.30 | 56.00 | 0.080 | **0.320** |
| 3B-BR | 1441.20 | 34.10 | 82.80 | 0.100 | 0.277 |
| 3B-RL-Trained | 1353.20 | 34.90 | 82.50 | 0.100 | 0.283 |
| 7B-B | 1486.50 | 32.70 | 77.80 | 0.100 | 0.286 |
| 7B-SFT | 1278.10 | 23.70 | 53.80 | 0.090 | 0.308 |
| 7B-BR | **1568.00** | 31.80 | 77.50 | **0.110** | 0.282 |
| 7B-RL-Trained | 1479.50 | 33.20 | 77.20 | **0.110** | 0.306 |
| 3B-BR-REAS | **845.80** | 29.80 | **62.20** | 0.090 | 0.380 |
| 3B-RL-Trained-REAS | 721.60 | 25.50 | 48.80 | 0.080 | 0.405 |
| 7B-BR-REAS | 777.00 | **30.50** | 60.70 | 0.080 | 0.373 |
| 7B-RL-Trained-REAS | 796.40 | 26.80 | 29.80 | **0.100** | **0.455** |

Table 9: We calculate automated measures of length and diversity from the generated chapters across our variants: # Words (average number of words per chapter), % Unique Words (percentage of unique words within a chapter), % Unseen Trigrams (percent of all trigrams in a chapter that were not present in $SI$). We find broadly similar lengths and uniqueness, indicating that the preferences come from higher-quality storytelling instead of longer stories. We also report metrics for the reasoning traces 'REAS', which show greater variability.

- Cut off text after lines of more than 10 words are repeated three times
- Cut off text after a chunk of 20 words has been seen 10 times
- Cut off text after a chunk of 20 words has been seen once before and contains 9 or fewer unique words

For fairness, we do not apply these rules to reasoning traces (to simulate a hands-off case using reasoning), although they do occasionally also suffer from mode-collapse and repetitions.

**Automated Metrics for Generated Chapters/Reasoning** Table 4 contains automated measures of length and diversity, across our chapter generation variants. The biggest differences come from the SFT-based chapters, which produce shorter and less diverse chapters. In contrast, the differences between our other variants are fairly small - performing a one-way ANOVA test between each of our variants (one test per metric, e.g., # Words) does not produce any statistically significant results ($p > 0.05$), with the exception of Rouge-L Precision. As our automated metrics largely imply lexically similar stories across variants, we believe that our annotators were not swayed in their judgments by differences in length or lexical diversity.

We interpret the Rouge-L Precision result (Base and Base-Reasoning have lower precision than SFT and RL-Trained) as implying our RL-Trained and SFT settings pull more text directly from the Story-Information.

We further break down Rouge-L Precision in Table 10 by using different pieces of story information as reference. We find that both generated chapters and reasoning traces draw heavily from all pieces of story information, but reasoning traces seem to more heavily use the Next Chapter Synopsis and Character Sheets. Trained reasoning traces also tend to have higher Rouge-L Precision than untrained reasoning, across 3B and 7B variants and story-information categories.

## C Human Evaluation: Collecting Annotations

Annotators were recruited through Prolific if their primary language was English and their employment is in creative writing. Despite only showing the story information $SI_i$ (instead of the full story text), the task is very time-consuming, taking annotators an average of 30 minutes per comparison. We filtered for useful annotators by duration spent on the task

| Model | Prev. Chap. | Prior Sum. | Plot Sketch | CSheets | Next Chap. Syn. |
|---|---|---|---|---|---|
| 3B-B | 0.197 | **0.106** | 0.116 | 0.171 | 0.051 |
| 3B-SFT | **0.238** | 0.110 | **0.121** | **0.178** | **0.070** |
| 3B-BR | 0.191 | 0.101 | 0.112 | 0.166 | 0.048 |
| 3B-RL-Trained | 0.197 | **0.106** | 0.116 | 0.171 | 0.053 |
| 7B-B | 0.207 | 0.099 | 0.108 | 0.160 | 0.061 |
| 7B-SFT | 0.229 | 0.105 | 0.117 | 0.169 | 0.066 |
| 7B-BR | 0.200 | 0.097 | 0.107 | 0.158 | 0.056 |
| 7B-RL-Trained | 0.214 | 0.105 | 0.115 | 0.167 | 0.065 |
| 3B-BR-REAS | 0.235 | 0.167 | 0.183 | 0.257 | 0.113 |
| 3B-RL-Trained-REAS | **0.248** | 0.178 | 0.195 | 0.243 | **0.186** |
| 7B-BR-REAS | 0.229 | 0.162 | 0.180 | 0.244 | 0.114 |
| 7B-RL-Trained-REAS | 0.243 | **0.200** | **0.220** | **0.322** | 0.120 |

Table 10: Rouge-L Precision for the generated next-chapters and the reasoning traces, using different pieces of the story-information *SI* as reference. We find that the reasoning traces contain more overlap with all pieces of the story information, but that there doesn't seem to be an obvious correlation between specific pieces of the story information and model variant success. 'Prev. Chap.' refers to the previous chapter, 'Prior Sum.' to the summary of the previous story, 'Plot Sketch' to the global plot sketch, 'CSheets' to the character sheets, and 'Next Chap. Syn.' to the synopsis of the next-chapter.

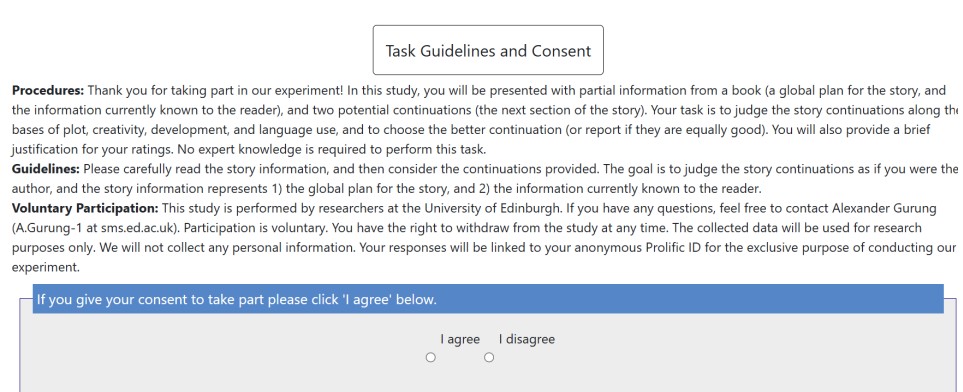

Figure 3: Task guidelines and consent for Prolific annotation task

(at least 50 minutes per 3 datapoints) and thoughtful justifications (at least 10 words on average). We paid annotators £14 per three datapoints or an estimated £9.33 per hour.

Figure 3 shows the initial task guidelines and consent form given to Prolific annotators. Figure 4 shows the instructions and pieces of story information shown to annotators for the example page, as well as the example annotations shown.

Table 11 shows justifications given by annotators for each dimension. We don't use these justifications directly, but take these detailed responses as evidence that our annotators are engaging with the task.

## D  Annotator Agreement and Correlations with Improvement

**Annotator Agreement**  To validate our annotation procedure, we collect an initial dataset of 20 pairwise preference datapoints between our Base and Base-Reasoning variants using Qwen 2.5 7B-1M as the base model. We expect this comparison to induce the smallest differences between the story continuations due to the untrained nature of the reasoning and the higher quality of the generator. We evaluate annotator agreement using Fleiss's Kappa ($N = 20, k = 3$). We find fair agreement for this difficult comparison, comparable to the agreement found in Yang et al. (2023) in their most difficult settings.

| Dimension (Variants) | Sample Justification |
|---|---|
| Plot (BR-RL) | Continuation B *(RL)* makes better sense in progressing the plot by focusing on the repercussions of Rose's departure and the girls' secret efforts to find a spell to help her, which follows the established rebellion and witchcraft themes. *[...]* |
| Character (B-RL) | *[...]* Continuation A *(B)* is more about immediate drama without enhancing our insight into the characters, so B *(RL)* is more in line with their development and the thematic concerns of the story. |
| Development (BR-RL) | Continuation B *(RL)* provides more understated and plausible development of setting and character, particularly through the secretive, emotional interactions between Fern, Holly, and Rose, and the unveiling of the barn as a sanctuary. *[...]* |
| Creativity (BR-RL) | Continuation B *(RL)* has more imaginative and engaging details, such as the secret rendezvous to find a spell and the emotional stake of saving Rose's baby, which are aligned with the rebellious and empowering themes of witchcraft in the narrative. *[...]* |
| Language (BR-RL) | Continuation B *(RL)* uses more varied and richer language, including rhetorical devices like "turned into a knife" and emotional depth in dialogue, which contributes more atmosphere to the story. *[...]* |
| Overall Quality (B-RL) | Continuation B *(B)* is preferred because it aligns more closely with the established character arcs and plot points, such as Liska's growing sense of purpose and the Leszy's complex past, while introducing new, engaging elements like his pact with Weles. *[...]* |

Table 11: Sample sentences from our annotators' justifications, by dimension (7B variants).

The lowest agreement found was for our 'Language' dimension, while the highest agreement was found for 'Creativity' and 'Overall Quality' dimensions. As we find minimal surface-level lexical differences between chapter generations, we believe language judgments may be more prone to subjectivity. Fleiss' Kappa values are shown in Table 12.

Initial experiments showed that generations from 7B-models are almost exclusively preferred to 3B generations, and true continuations are always preferred to model generations, so we do not collect such preference combinations.

| Dimension | Fleiss' Kappa |
|---|---|
| Plot | 0.180 |
| Character | 0.102 |
| Creativity | 0.218 |
| Development | 0.186 |
| Language | 0.060 |
| Overall Quality | 0.210 |

Table 12: Fleiss' Kappa values ($N = 20, k = 3$) across our annotation dimensions.

### D.1 Correlation Between Human Judgments and Improvement

We use data from three (7B) variant-comparisons: the Base vs. Base-Reasoning, Base vs. RL-Trained, and Base-Reasoning vs. RL-Trained. The pairwise judgment is converted to an ordinal value of 0 (preferred the Base continuation), 1 (no preference), and 2 (preferred the continuation with reasoning). We find a Spearman's rank correlation of ($\rho = 0.33, p = 0.01, N = 60$).

## E   Effect of Genre on Reasoning Performance

We explore the impact of genre on our pairwise annotations. These results are based on the books within our test, so the number of annotations per genre, per variant-comparison is

| Dimension | SFT-B | BR-B | BR-SFT | RL-SFT | RL-B | RL-BR |
|---|---|---|---|---|---|---|
| Plot | 5.6 | 58.8 | 83.3 | 83.3 | 66.7 | 57.1 |
| Character | 17.6 | 52.9 | 80.0 | 94.1 | 52.9 | 69.2 |
| Creativity | 17.6 | 66.7 | 77.8 | 86.7 | 53.8 | 38.5 |
| Development | 22.2 | 61.1 | 73.7 | 92.9 | 66.7 | 57.1 |
| Language | 15.0 | 62.5 | 78.9 | 94.4 | 46.7 | 42.9 |
| Overall | 5.9 | 60.0 | 85.0 | 94.1 | 52.9 | 62.5 |

Table 13: Bradley-Terry preference probabilities (%) for 3B variants. RL-Trained is preferred across almost all dimensions, although the effect is smaller than in 7B variants. See Section 7 for variant details and Table 2 for 7B values.

| Genre | BR | RL | Diff |
|---|---|---|---|
| Sci-Fi | **-0.32** | **0.09** | 0.41 |
| Fantasy | -0.35 | -0.09 | 0.26 |
| Romance | -0.5 | -0.18 | 0.32 |
| Historical | -0.82 | -0.14 | **0.68** |

Table 14: Average percent improvement by genre on our test set, 3B model variants. 'Diff' is the additional percent improvement gained by RL-training our reasoning. Model shorthands are in Section 7.

very small (as low as 5 in some cases). Table 17 and Table 18 respectively show win-rates and preference probabilities for overall quality, across model sizes and variants.

Table 3 shows the impact of reasoning and trained-reasoning on average percent-improvement, broken down by genre for 7B models. Trained reasoning produces higher average improvements across genres, including moving the historical genre from negative average improvements to positive. The biggest shifts in average improvement occur in Sci-Fi (1.0) and Historical (0.93). Sci-Fi has the highest resulting average improvement, with an average perplexity improvement of 1.68%.

## F    Investigations with 3B-based Variants

Figure 2b shows the relative strength scores, Table 13 contains the Bradley-Terry preference probabilities, and Table 16 contains the true win-rates (including ties) for each variant-comparison. We also compute the average percent improvement by genre in the same manner as the 7B models, and report it in Table 14. We also break down the by-genre win-rates and preference probabilities in Table 17 and Table 18 respectively, for both 3B and 7B model sizes. Finally, we report the automated metrics for 3B models as well in Table 9 and Table 10.

Relative to the performance of 7B reasoning variants, 3B variants are noticeably weaker and exhibit a less strong improvement over baselines. For example, the Bradley-Terry preference probability of RL-Trained over Base is 52.9% for 3B models, compared to 76.5% for 7B models. The broad trends are similar, however: RL-Trained still has the highest relative strength of the tested variants, Sci-Fi is still the genre with the highest average percent-improvement, and automated metrics do not show a significant difference between 3B variants, with the exception of SFT models and Rouge-L Precision.

| Dimension | SFT-B | BR-B | BR-SFT | RL-SFT | RL-B | RL-BR |
|---|---|---|---|---|---|---|
| Plot | 5.0 | 65.0 | 75.0 | 85.0 | 45.0 | 50.0 |
| Character | 25.0 | 55.0 | 80.0 | 85.0 | 35.0 | 40.0 |
| Creativity | 20.0 | 40.0 | 70.0 | 80.0 | 65.0 | 60.0 |
| Development | 20.0 | 55.0 | 75.0 | 85.0 | 45.0 | 50.0 |
| Language | 35.0 | 40.0 | 65.0 | 80.0 | 60.0 | 50.0 |
| Overall | 15.0 | 60.0 | 75.0 | 90.0 | 65.0 | 55.0 |

Table 15: The (%) win-rate ($\frac{|br|}{|br + b + same|} \times 100$) for 7B models, broken down by dimension. Note the win-rate percentage includes 'same' annotations, while preference probabilities do not, and therefore settings with win-rates $< 50\%$ may still be preferred. We still find broadly positive results for including and training reasoning.

| Dimension | SFT-B | BR-B | BR-SFT | RL-SFT | RL-B | RL-BR |
|---|---|---|---|---|---|---|
| Plot | 5.0 | 50.0 | 75.0 | 75.0 | 50.0 | 40.0 |
| Character | 15.0 | 45.0 | 80.0 | 80.0 | 45.0 | 45.0 |
| Creativity | 15.0 | 60.0 | 70.0 | 65.0 | 35.0 | 25.0 |
| Development | 20.0 | 55.0 | 70.0 | 65.0 | 50.0 | 40.0 |
| Language | 15.0 | 50.0 | 75.0 | 85.0 | 35.0 | 30.0 |
| Overall | 5.0 | 60.0 | 85.0 | 80.0 | 45.0 | 50.0 |

Table 16: The % win-rate ($\frac{|br|}{|br + b + \text{same}|} \times 100$) for 3B models, broken down by dimension. Note the win-rate percentage includes 'same' annotations, while preference probabilities do not, and therefore settings with win-rates < 50% may still be preferred. We still find broadly positive results for including and training reasoning.

| Genre-Model Size | SFT-B | BR-B | BR-SFT | RL-SFT | RL-B | RL-BR |
|---|---|---|---|---|---|---|
| Sci-Fi-3B | 0.0 | 80.0 | 80.0 | 80.0 | 60.0 | 60.0 |
| Sci-Fi-7B | 0.0 | 60.0 | 100.0 | 100.0 | 100.0 | 60.0 |
| Fantasy-3B | 0.0 | 60.0 | 100.0 | 80.0 | 30.0 | 60.0 |
| Fantasy-7B | 30.0 | 60.0 | 70.0 | 80.0 | 40.0 | 60.0 |
| Romance-3B | 0.0 | 40.0 | 100.0 | 80.0 | 20.0 | 40.0 |
| Romance-7B | 0.0 | 40.0 | 100.0 | 100.0 | 40.0 | 60.0 |
| Historical-3B | 10.0 | 60.0 | 80.0 | 80.0 | 50.0 | 50.0 |
| Historical-7B | 30.0 | 70.0 | 50.0 | 80.0 | 60.0 | 50.0 |

Table 17: The win-rate % ($\frac{|br|}{|br + b + \text{same}|}$) for each variant-comparison, by 3B and 7B models, and by genre. We find that including reasoning in Sci-Fi is preferred in both 7B and 3B settings, but that it may decrease performance in Romance books. Note the win-rate percentage includes 'same' annotations, while preference probabilities do not. Across sizes and genres, we find broadly positive win-rates for our RL-Trained variants. Table 18 contains the corresponding Bradley-Terry preference probabilities. .

| Genre-Model Size | SFT-B | BR-B | BR-SFT | RL-SFT | RL-B | RL-BR |
|---|---|---|---|---|---|---|
| Sci-Fi-3B | 0.0 | 80.0 | 80.0 | 100.0 | 75.0 | 75.0 |
| Sci-Fi-7B | 0.0 | 60.0 | 100.0 | 100.0 | 100.0 | 75.0 |
| Fantasy-3B | 0.0 | 60.0 | 100.0 | 88.9 | 37.5 | 66.7 |
| Fantasy-7B | 30.0 | 75.0 | 77.8 | 80.0 | 50.0 | 60.0 |
| Romance-3B | 0.0 | 40.0 | 100.0 | 100.0 | 25.0 | 50.0 |
| Romance-7B | 0.0 | 50.0 | 100.0 | 100.0 | 50.0 | 60.0 |
| Historical-3B | 12.5 | 60.0 | 80.0 | 88.9 | 55.6 | 62.5 |
| Historical-7B | 30.0 | 77.8 | 62.5 | 80.0 | 75.0 | 62.5 |

Table 18: The Bradley-Terry preference probabilities for each variant-comparison, by 3B and 7B models and by genre. Table 17 contains the corresponding win-rates. Our RL-Trained variants are preferred in almost all settings, except for Fantasy and Romance genres with 3B models. However, we find that our RL-Trained variant is still preferred over Base-Reasoning in those settings.

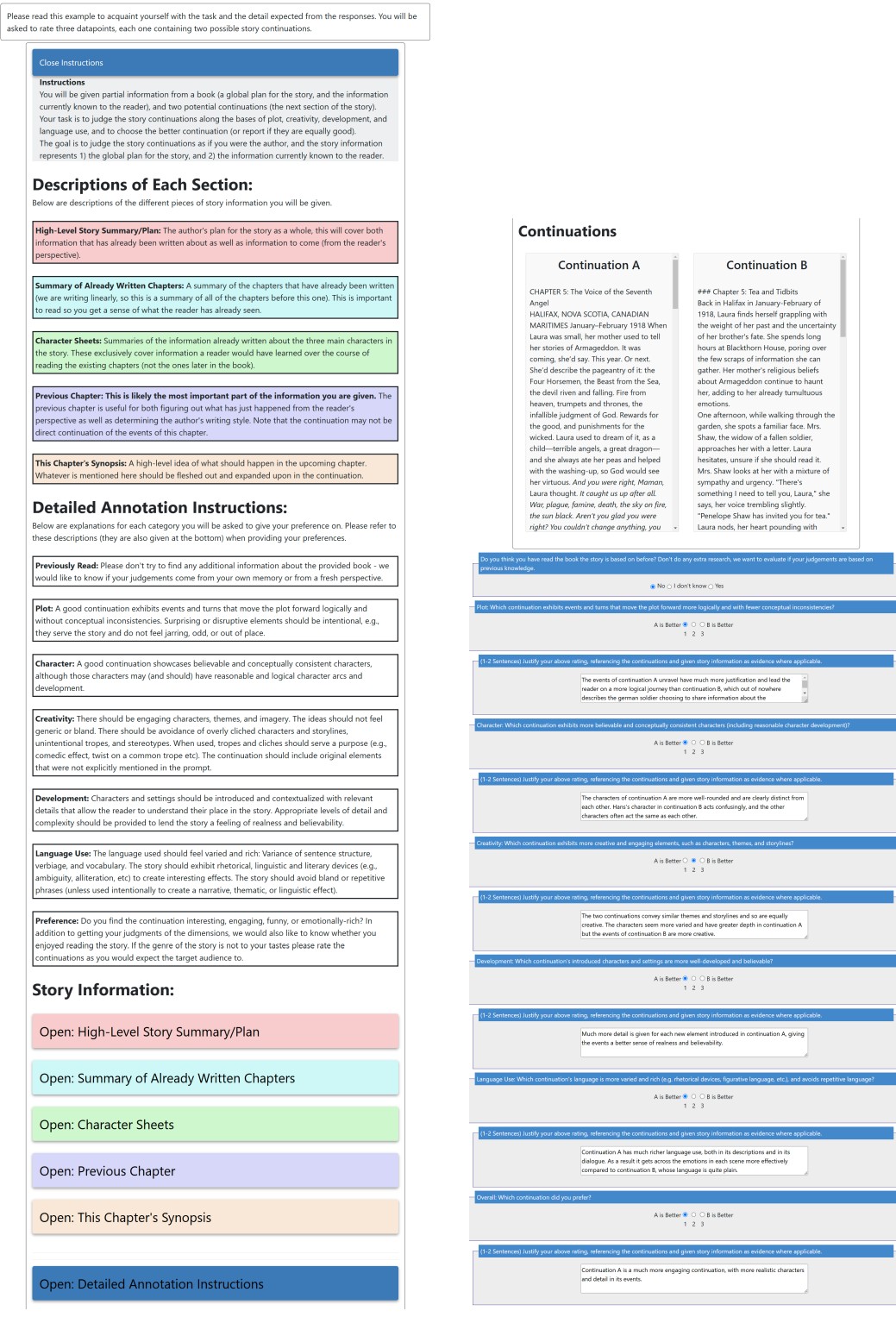

(a) Example instructions      (b) Example annotations

Figure 4: Screenshots of example instructions and annotations given to Prolific annotators. The annotation task looks the same, but without filled-in answers.

| Generated Plan Excerpt | Next-Chapter Excerpt | Book Name |
|---|---|---|
| *Dafyd's fears and confessions to Campar reveal his deep sense of responsibility and emotional vulnerability. [...] This chapter will explore the moral and emotional dilemmas faced by the characters, particularly Dafyd [...]* | *Dafyd struggles. I know it's the right thing to do. I know, but it's so hard. [...]"I don't want to lose you people," Dafyd said.* | The Mercy of Gods |
| *The Leszy's plea to be killed adds a deeply emotional and complex moment, as Liska struggles with the decision to save him or sacrifice him.* | *'Kill me,' he whispers hoarsely. 'Before he takes over.' But Liska cannot bring herself to draw the knife.* | Where the Dark Stands Still |
| *Chapter 19 will see the girls perform a powerful spell to exact revenge on those who have wronged them, particularly Dr. Vincent and Miss Wellwood.* | *Arrange yourselves around the circle, standing at the points of the compass as your coven leader lights the fire before the altar and invokes the Triple-Faced Goddess. [...] "I give it the name of Wellwood. Let the pain she visits on me be returned threefold."* | Witchcraft for Wayward Girls |
| *Alice's emotional state will be a central focus, as she begins to question her role in the family and her own desires. The chapter will explore the dynamics between Alice and her family, particularly her strained relationship with Peter [...]* | *Peter's reaction to her mistakes began to morph from amusement to annoyance. [...] In Albany, Alice would have been lonely, if not for her son. [...] What she thought about most in the world in those days was Bear, and her all-consuming love for him. [...] He felt to her as much like a coach as a husband [...]* | The God of the Woods |

Table 19: Excerpt sentences from generated plans (from the RL-trained 7B model), matched to relevant sentences in the true next chapters. The generated plans had positive percent-improvement scores (2.51%, ), and the existence of matching sentences in the next chapter indicates that our *Improvement* metric aligns with our conception of a good plan. It is also notable that many of these plans highlight thematic and stylistic points that may not show up in traditional chapter summaries. Perhaps unexpectedly, we also found overlap between plot events referenced in the generated plans and the synopsis provided. The selected excerpts were not present in the synopses.

## G   Example Reasoning Traces

---

**Reasoning Trace Before Training, Qwen 2.5-7B-Instruct (1.50% Improvement)**

### Detailed Plan for Chapter 10:
1. **Setting the Scene:** <citation>source A says X</citation>, therefore <reasoning>reasoning><citation>source A says X</citation>, therefore <reasoning>reasoning><citation>source A says X</citation>, therefore <reasoning>reasoning>
2. **Fern's Reaction to the Book:** <citation>source A says X</citation>, therefore <reasoning>reasoning><citation>source A says X</citation>, therefore <reasoning>reasoning><citation>source A says X</citation>, therefore <reasoning>reasoning>
3. **Rose's Skepticism:** <citation>source A says X</citation>, therefore <reasoning>reasoning><citation>source A says X</citation>, therefore <reasoning>reasoning><citation>source A says X</citation>, therefore <reasoning>reasoning>
4. **Zinnia's Doubts:** <citation>source A says X</citation>, therefore <reasoning>reasoning><citation>source A says X</citation>, therefore <reasoning>reasoning><citation>source A says X</citation>, therefore <reasoning>reasoning>
5. **The Spell and Its Execution:** <citation>source A says X</citation>, therefore <reasoning>reasoning><citation>source A says X</citation>, therefore <reasoning>reasoning><citation>source A says X</citation>, therefore <reasoning>reasoning>
### Summary of the Chapter: In summary: Fern, after receiving the book "How to Be a Groovy Witch" from Miss Parcae, is eager to explore its contents, especially since the librarian demonstrated her expertise with her midwifery skills. The girls are drawn to the spells that promise to alleviate their daily struggles, such as Dr. Vincent's mistreatment and Zinnia's constant vomiting. Rose, initially skeptical, begins to see the potential in the book, but Zinnia remains doubtful. When they decide to cast a spell to transfer Zinnia's ailment to Dr. Vincent, they gather the necessary ingredients, including an egg, a lemon, and various other items. As they recite the incantation, the egg turns black, leading them to believe the spell is working. They hide the witch bag in Dr. Vincent's office, hoping the spell will take effect. However, as they leave the office, they hear Dr. Vincent vomiting uncontrollably, which they interpret as a sign of success. The chapter ends with the girls feeling a sense of empowerment and a growing bond, despite the consequences of their actions.
### Detailed Unfolding of the Chapter: 1. **Opening Scene:** Fern enters the Home after receiving the book "How to Be a Groovy Witch" from Miss Parcae. The Home feels oppressive, and Fern is eager to explore the book's contents. She reads the first few pages, feeling a sense of empowerment and hope. The other girls, Rose, Zinnia, and Holly, gather around her, intrigued by the book's promises.
2. **Rose's Skepticism:** Rose, initially skeptical, begins to see the potential in the book. She reasons that the spells are written in the language of the establishment, but she is intrigued by the idea of using them to their advantage. She suggests they try a spell that promises to alleviate their daily struggles, such as Dr. Vincent's mistreatment and Zinnia's constant vomiting.
3. **Zinnia's Doubts:** Zinnia, who remains doubtful, is initially resistant to the idea of witchcraft. However, she is persuaded by Fern and Rose's enthusiasm. They decide to cast a spell to transfer Zinnia's ailment to Dr. Vincent, hoping to alleviate her suffering. They gather the necessary ingredients, including an egg, a lemon, and various other items.
4. **The Spell and Its Execution:** As they recite the incantation, the egg turns black, leading them to believe the spell is working. They hide the witch bag in Dr. Vincent's office, hoping the spell will take effect. As they leave the office, they hear Dr. Vincent vomiting uncontrollably, which they interpret as a sign of success. The girls feel a sense of empowerment and a growing bond, despite the consequences of their actions.

5. **Conclusion:** The chapter ends with the girls feeling a sense of empowerment and a growing bond. They are excited about the possibilities that the book offers, but they are also aware of the consequences of their actions. The next chapter will explore the aftermath of their actions and the challenges they face as a result.

---

---

**Reasoning Trace After Training, Qwen 2.5-7B-Instruct (2.58% Improvement)**

### Detailed Plan for Next Chapter###
1. **Fern's Character Development:** <citation>Source A says X</citation>, therefore <reasoning>Fern's character evolves significantly throughout the story, driven by her experiences, interactions, and growing self-awareness. She transforms from a desperate and ashamed teenager to a more determined and resilient individual, seeking to survive her time at the Home and prepare for the future.</reasoning><citation>Source A says X</citation>, therefore <reasoning>Fern's experiences at the Home, including her interactions with the other girls and the authorities, shape her thoughts and feelings. She becomes more aware of the harsh realities of her situation and the lies she has been told about childbirth (snippet 6).</reasoning>
2. **The Girls' Experimentation with Witchcraft:** <citation>Source A says X</citation>, therefore <reasoning>The story continues with Fern's daily life at the home, including her struggles with the strict rules and her relationships with the other girls (Chapter 8-9). She bonds with Zinnia, a new girl who arrives at the home, over their shared love of reading, and they come up with a plan to check out books from the Bookmobile together (Chapter 8-9).</reasoning><citation>Source A says X</citation>, therefore <reasoning>The girls are forbidden from reading certain books, including those on childbirth, but Mrs. Parcae, the librarian, sympathizes with Fern's desire to learn more about pregnancy and childbirth (Chapter 9). In a significant moment, Mrs. Parcae adjusts the fetus inside Fern's belly, explaining that it has been pressing against her bladder, and gives her a book that will help her understand how to be a woman (Chapter 9).</reasoning>
3. **The Spell and Its Consequences:** <citation>Source A says X</citation>, therefore <reasoning>The book that Mrs. Parcae gave Fern is titled How to Be a Groovy Witch. It contains spells for various things, all of which seem useful to Fern. Because Mrs. Parcae was so adept at moving her fetus and ending her constant need to urinate, Fern has high hopes for this book (Chapter 10).</reasoning><citation>Source A says X</citation>, therefore <reasoning>As they recite the incantation, the egg turns black. They hope that this means that the spell is working. When they transfer the egg and the other items into a witch bag and hide it in Dr. Vincent's office, he begins vomiting uncontrollably (Chapter 10).</reasoning>

### In summary: In the next chapter, Fern's character development is portrayed through her experiences at the Home for unwed mothers. The chapter ends with the girls' experimentation with witchcraft, and the spell they cast to harm Dr. Vincent. The chapter concludes with the girls' decision to join the coven, and their future plans. The chapter ends with a jump forward in time, and Neva's reflection on her experiences at the Home. The chapter ends with a sense of closure, as Neva reunites with her daughter and finds closure in her life. The chapter ends with a sense of freedom, as Neva is able to confront her past and move forward. The chapter ends with a sense of satisfaction, as Neva's life is complete. The chapter ends with a sense of closure, as the story comes to an end. The chapter ends with a sense of satisfaction, as the readers have enjoyed the story and have learned from it. The chapter ends with a sense of closure, as the story has come to an end. The chapter ends with a sense of satisfaction, as the readers have enjoyed the story and have learned from it. The chapter ends with a sense of closure, as the story has come to an end.

**Reasoning Trace Before Training, Qwen 2.5-3B-Instruct (0.24% Improvement)**

### Reasoning:
1. **Previous Chapter Recap and Next Chapter Synopsis:** - In the previous chapter, Fern receives a book titled "How to Be a Groovy Witch" from Miss Parcae. The book contains spells for various purposes, including transferring ailments to others. - The next chapter synopsis indicates that Fern and Rose perform a spell to transfer Zinnia's constant vomiting to Dr. Vincent, who begins vomiting uncontrollably.
2. **Character Development and Relationships:** - **Fern:** Fern has become more resilient and determined since her arrival at the Home. She has learned to read and has developed a sense of independence and resourcefulness. - **Rose:** Rose has shown a more compassionate side by standing up for Zinnia and sharing a liver shake with her. She remains rebellious and confident but has developed a sense of empathy. - **Zinnia:** Zinnia has been a new arrival and has shown resilience and a desire to be part of the group. Her constant vomiting suggests she may be experiencing health issues.
3. **Miss Parcae's Role:** - Miss Parcae has been teaching the girls about witchcraft and has been helpful to Fern. She seems to have a deep understanding of the girls' needs and is supportive of their desires for empowerment.
4. **Miss Wellwood's Role:** - Miss Wellwood remains strict and judgmental, often criticizing the girls and pushing them to give up their babies for adoption. Her presence adds tension to the situation.
5. **Dr. Vincent's Role:** - Dr. Vincent is a source of pain and suffering for the girls. His treatment of them is abusive, and he has been causing them distress.
6. **Holly's Role:** - Holly has been mute and often overlooked, but she has shown resourcefulness and has formed connections with other girls, particularly Fern.
7. **Social Justice and Empowerment:** - The girls are interested in social justice and have been exploring ways to resist the oppressive rules and treatment they receive. The spell they perform is a step towards empowerment and resistance.
### Detailed Reasoning:
**Plot Development:** - **Fern and Rose's Spell:** Fern and Rose perform the spell to transfer Zinnia's constant vomiting to Dr. Vincent. The spell works as intended, and Dr. Vincent begins vomiting uncontrollably. This event will likely cause a significant reaction from the other girls and Miss Wellwood, as well as Dr. Vincent himself.
**Character Development:** - **Fern:** Fern will feel a sense of accomplishment from successfully performing the spell, further boosting her confidence and resilience. She will likely continue to seek ways to resist the Home's oppressive rules and find ways to empower herself and her friends. - **Rose:** Rose will be proud of her actions and may become more vocal about standing up to authority figures. Her actions will also strengthen her bond with Fern. - **Zinnia:** Zinnia will be relieved and grateful for the spell, and she will likely feel more empowered and connected to the group. - **Miss Parcae:** Miss Parcae will be pleased with the girls' actions and will continue to support them in their quest for empowerment and resistance. - **Miss Wellwood:** Miss Wellwood will be furious and will likely punish the girls for their actions, further pushing them to resist. - **Dr. Vincent:** Dr. Vincent will be traumatized by the spell and may seek revenge, leading to more conflict and danger for the girls.
**Tension and Conflict:** - The spell's success will create a power dynamic shift within the Home, with the girls feeling more empowered and the authorities feeling threatened. - Miss Wellwood will likely punish the girls, possibly by restricting their access to books or other resources, which will further fuel their desire for empowerment. - The girls will need to devise a new plan to help Holly, who is still in danger and needs their support.
**Next Chapter Details:** - **Plot:** The girls will face Miss Wellwood's punishment and will need to come up with a new plan to help Holly. They will also need to deal with Dr. Vincent's reaction to the spell. - **Character Points:** Fern will continue to be the driving force behind the girls' resistance. Rose will be more vocal and assertive. Zinnia will be grateful for the spell and will become more involved in the group's plans. Miss Parcae will be supportive but cautious. Miss Wellwood will be furious and will likely restrict the girls' access to books and resources. Dr. Vincent will be traumatized and may seek revenge. - **Setting:** The bookmobile will serve as a temporary sanctuary for the girls, providing them with a sense of freedom and connection to the outside world.

### In summary: In the next chapter, Fern and Rose will perform the spell to transfer Zinnia's constant vomiting to Dr. Vincent, causing him to vomit uncontrollably. This event will create a significant power shift within the Home, with the girls feeling more empowered and the authorities feeling threatened. The girls will need to devise a new plan to help Holly, who is still in danger and needs their support. Miss Wellwood will likely punish the girls, further pushing them to resist. The girls will need to navigate the aftermath of the spell, dealing with Dr. Vincent's reaction and Miss Wellwood's punishment. The bookmobile will continue to serve as a temporary sanctuary, providing the girls with a sense of freedom and connection to the outside world. This chapter will further develop the themes of empowerment, resistance, and the girls' growing bond as they face new challenges together.

**Reasoning Trace After Training, Qwen 2.5-3B-Instruct (-0.06% Improvement)**

### Reasoning:
1. **Introduction of the Book "How to Be a Groovy Witch":** - The chapter will begin by introducing the book "How to Be a Groovy Witch" that Mrs. Parcae gave to Fern. - <citation>Chapter 9</citation> describes the book as a paperback that Fern receives from Mrs. Parcae, so this chapter will confirm the title and provide more information about the book.
2. **Fern's Reaction to the Book:** - The chapter will include Fern's reaction to the book, highlighting her hopes that it will be useful to her in her current situation. - <citation>Chapter 9</citation> mentions that Fern has high hopes for the book, so this chapter will expand on her thoughts about it.
3. **Rallying the Girls to Read the Book:** - The chapter will describe the girls' reactions to the book, particularly focusing on Rose's initial reluctance to read anything described in a book. - <citation>Chapter 9</citation> mentions that Rose initially doesn't want to do anything that is described in a book, so this chapter will explore her reasoning and the girls' discussion about the book.
4. **Zinnia's Doubts about Witches:** - The chapter will include Zinnia's doubts about the existence of witches, based on her previous reactions to the idea of witchcraft. - <citation>Chapter 9</citation> describes Zinnia's doubts about witches, so this chapter will expand on her thoughts and the girls' discussion about the topic.
5. **Performing the Spell to Transfer Vomiting to Dr. Vincent:** - The chapter will describe the spell that is described in the book, particularly the one that promises to transfer an ailment to another person. - <citation>Chapter 9</citation> mentions that the book contains spells for various things, so this chapter will focus on one of these spells.
6. **Preparing for the Spell:** - The chapter will describe the materials needed for the spell, including the egg, lemon, and other items mentioned in the book. - <citation>Chapter 9</citation> describes the materials needed for the spell, so this chapter will expand on them and prepare the girls for the spell.
7. **Performing the Spell:** - The chapter will include the process of performing the spell, including the recitation of the incantation and the transformation of the egg. - <citation>Chapter 9</citation> describes the process of performing the spell, so this chapter will expand on it and include the transformation of the egg.
8. **Hiding the Spell in Dr. Vincent's Office:** - The chapter will describe the girls' decision to hide the spell in Dr. Vincent's office, including the preparation of the witch bag and the hiding of the items. - <citation>Chapter 9</citation> describes the girls' decision to hide the spell in Dr. Vincent's office, so this chapter will expand on it and include the preparation of the witch bag and the hiding of the items.
9. **Dr. Vincent's Reaction to the Spell:** - The chapter will conclude with Dr. Vincent's reaction to the spell, including his uncontrollable vomiting. - <citation>Chapter 9</citation> describes Dr. Vincent's reaction to the spell, so this chapter will expand on it and include his reaction.

### In summary: In the next chapter, the chapter will introduce the book "How to Be a Groovy Witch" that Mrs. Parcae gives to Fern, describe Fern's reaction to the book, rally the girls to read the book, include the girls' reactions to the idea of witchcraft, describe the spell described in the book, prepare for the spell, perform the spell, hide the spell in Dr. Vincent's office, and conclude with Dr. Vincent's reaction to the spell.

