# OpenReview forum: "Learning to Reason for Long-Form Story Generation"
_colmweb.org/COLM/2025/Conference — COLM 2025_

### Official Review · Reviewer_uAHM · 2025-04-12

**Rating:** 6
**Confidence:** 3
**Ethics Flag:** 1

**Summary:**

This paper proposes a reasoning-enhanced framework for long-form story generation, where the generation process is divided into three phases: reasoning, planning, and writing. The authors introduce a task called Next-Chapter Prediction (NCP) and design a verifiable reward function (VR-CLI) based on perplexity improvement to train the reasoning component using reinforcement learning. Experiments across multiple models and genres demonstrate the effectiveness of this approach in improving story quality.

**Reasons To Accept:**

1. The paper frames story generation as a structured process of “reasoning → planning → writing,” which aligns more closely with how human authors compose stories. This formulation is more principled than traditional end-to-end generation approaches.

2. The proposed reward signal, comparing generation likelihood with and without reasoning traces, does not require any labeled data and is intuitively sound.

3. The paper is clearly written and easy to follow.

**Reasons To Reject:**

The main concern lies in whether perplexity improvement is truly a suitable proxy for story quality in the VR-CLI reward formulation. The paper currently lacks an intuitive explanation for why a decrease in PPL should correlate with a better story. It would help to include more analysis or concrete examples to justify the reasonableness of this proxy reward.

+++++ update+++++

Further, I examined the section titled “Perplexity improvement correlates with human judgements.” The reported Spearman correlation (ρ = 0.33, p = 0.01) suggests a moderate positive relationship, indicating that higher-quality reasoning traces tend to yield lower PPL. However, I couldn’t find a clear definition of what constitutes “generation quality” in this context. Does a higher Improvement reflect better fluency, coherence, or something else? This should be clarified explicitly.

Despite my concern about the use of perplexity as a reward signal, I still find the idea of leveraging unsupervised signals from the story generation process itself for RL both promising and scalable, as such data can be generated in large quantities.

---

> ### Author Response · Authors · 2025-05-28
>
> Thank you for your review! We are glad you found our formulation principled and hope to explain our intuition behind our use of perplexity:
>
> Addressing Concerns:
>
> “[...] whether perplexity improvement is truly a suitable proxy for story quality in the VR-CLI reward formulation”
>
> Our explanation of our intuition behind perplexity improvement as a reward is in our response to all reviewers (1), but we will elaborate on your other points here to answer your concerns more directly.
>
> While our intuition is that plans with higher percent-improvement contain more accurate information and fewer distractions, we do not evaluate their quality directly as they are not the desired end-product. Instead, we evaluate our generated plans via the chapters they help generate. These chapters are evaluated by pairwise comparisons across the dimensions described in Section 7.2 (Plot, Character, Creativity, Development, Language, and Overall). Figure 2 shows the relative strengths of each variant broken down across these dimensions - we find that our reward and training methodology improves judgements across all metrics for 7B-models and the majority of metrics for 3B.
>
> In the section you highlighted titled “Perplexity improvement correlates with human judgements”, we focus on validating our key assumption (on lines 318-324) that plans with higher percent-improvement will produce preferred chapters, by computing the correlation between perplexity improvement and the pairwise “overall preference” (line 282). Full annotation instructions are in Figure 4. The significant correlation we find implies that plans which induced likelihood improvements for the gold chapter, produced preferred story continuations.
>
> As mentioned in our response to all reviewers we can add clarifications and example annotated plans to demonstrate how the plans may relate to the increase/decreases in perplexity.

---

> > ### Comment · Reviewer_uAHM · 2025-06-01
> >
> > Thank you for your response. The general and individual response have partially solved my concerns.
> >
> >  best，

---

> > > ### Author Response · Authors · 2025-06-01
> > >
> > > Thank you for your response! If there are any other concerns we would be happy to address them, and if we addressed your concerns we hope you would consider taking that into account in your score.

---

### Official Review · Reviewer_RQpG · 2025-04-26

**Rating:** 7
**Confidence:** 3
**Ethics Flag:** 1

**Summary:**

This paper tackles the challenging task of long-form story generation with large language models (LLMs), which requires maintaining coherence, character development, and stylistic consistency across thousands of tokens. The authors introduce a novel reinforcement learning (RL) framework—Verifiable Rewards via Completion Likelihood Improvement (VR-CLI)—to guide a reasoning model in generating structured plans for upcoming story chapters. The experiments validate the superiority of the proposed model.

**Questions To Authors:**

1. How sensitive is the model performance to the threshold values in Eq. 7? Have you tried treating this as a continuous (rather than piecewise) reward signal? Have you conducted any ablation studies to assess how variations in these thresholds affect model convergence or final story quality?
2. Have you considered evaluating against stronger RL baselines beyond non-trained or SFT models? If not, what are the practical challenges in integrating, for example, preference-based RL methods like DPO?

**Reasons To Accept:**

1. Long-form story generation remains one of the most demanding applications for LLMs, where challenges like maintaining global coherence, avoiding repetition, and character consistency are magnified. This paper addresses these directly by proposing a task (NCP) that explicitly integrates story structure.
2. The use of verifiable rewards via completion likelihood improvement (VR-CLI) is an original contribution. It avoids the need for hand-labeled rewards or hard-to-define heuristics by instead relying on a measurable and model-driven improvement signal. This strikes a good balance between tractability and expressiveness for creative writing.
3. Leveraging a dataset of real-world books (with chapter summaries, character sheets, etc.) makes the method practical and scalable. The method for constructing structured story information from raw text is well-motivated and carefully documented. The experiments show the effectiveness of the proposed model.

**Reasons To Reject:**

1. While the authors compare their method against several baselines (e.g., SFT, non-trained reasoning), more state-of-the-art RL-based methods could strengthen the comparison.
2. The reward thresholding in Eq. 7 is not well-justified. The current formulation uses manually selected cutoffs (e.g., 0.05, 1, 2), which could significantly affect training dynamics. The paper would benefit from an ablation or sensitivity analysis exploring how these thresholds influence performance. Moreover, it would help to ground the thresholds in empirical observation or provide theoretical rationale.

---

> ### Author Response · Authors · 2025-05-28
>
> Thank you for your review! We appreciate the highlighting of our task’s difficulty and the practicality of our method. We address your concerns in the response to all reviewers, we point to the relevant sections per-concern below.
>
> Addressing concerns:
> R1: “[...] more state-of-the-art RL-based methods could strengthen the comparison”
> See our response to all reviewers (2), where we discuss why other RL approaches were not used.
>
> R2: Justifying Reward Thresholds
> See our response to all reviewers, (1).
>
> Answering Questions:
>
> Q1: Sensitivity to Reward Thresholds
> See our response to all reviewers, (1, especially the first two paragraphs).
>
> Q2: Have We Considered Other RL Baselines?
> See our response to all reviewers (2, especially the last paragraph).

---

> > ### Comment · Reviewer_RQpG · 2025-05-31
> >
> > I appreciate the author response and can understand the reasons not to compare with other RL baselines.

---

### Official Review · Reviewer_oa9L · 2025-05-13

**Rating:** 7
**Confidence:** 4
**Ethics Flag:** 1

**Summary:**

The authors present an RL based training method for long-form story generation. They fine-tune a Qwen2.5 model on a newly proposed task of next chapter prediction (NCP), where, instead of direct prediction of book-length generations, they generate one chapter after the other (with chapters predicted so far being provided as context). The intuition is that the reasoning generated by the model will improve the overall story compared to an SFT that quickly overfits. They use GRPO for training and for the reward they use improvement in per-token perplexity (PPL). Specifically, the reward model is thresholded piece-wise function with the actual reward values depending on the improvement in PPL. This reward paradigm is referred to as VR-CLI in the paper, or, verifiable rewards via completion likelihood improvement, and it is inspired by prior work RLVR (RL via verifiable rewards).

They curate a dataset of 30 books covering four genres: scifi, fantasy, romance, and historical; and they use 22 of them to train their models. They conduct quantitative and qualitative evaluation, where they compare their RL trained model with multiple baselines including an SFT and no RL baseline. Both quantitative and qualitative results show that the proposed VR-CLI framework leads to better stories compared to the baselines and is favored by humans (although human written stories are still preferred almost exclusively over all).

Overall, I like the experimental setup and this work has some merit. It shows that RL can work for a creative task while being more effective compared to SFT. My main concerns revolve around using PPL as a target for a creative task and potential overtuning of reward values (meaning that if someone were to do this for their own dataset for the same task they would likely need to spend a lot of time figuring the rewards values out) as the authors cannot release the dataset nor the models due to copyright issues.

**Questions To Authors:**

- Please add Table 6 and 7 (or a compressed version of them) to the main paper. It's an important detail showcasing how long is long really in the context of this work. I know you mention it in Section 4 but a table is always a good redundancy for such key details.
- What other rewards have you tried (that "didn't work")?
- Have you tried to use anything other than PPL for your reward? Coming up with those threshold values would've been surely painful, and since you already have a good sense of what metrics to look for (as you ask during human eval), have you considered using something like G-Eval scores (with an open-source base model of course) to provide you with the rewards?
- Table 3 shows mean percent improvement of what exactly? And what is the range there, 0-1 or 0-100? And what is the base model (the 0 model)?
- Why specifically chapter-level? Have you tried more finer or coarser-level of planning?
- Re 3B v/s 7B models, I think it is important to consider not just the number of parameters, but also the data the base model has been exposed to. For instance, your finding may not hold for Qwen3-4b, which is better than even Qwen2.5-72b on some tasks (but since I know Qwen3 has been exposed to more data, this 4b v/s 72b comparison is not really fair).

**Reasons To Accept:**

- The paper presents a novel take at long-form story generation using the task of next chapter prediction with RL.
- The presented framework is promising as it makes RL work for long text generation task (whereas RL has traditionally known to be notoriously difficult to stabilize for NLP tasks)
- The reward modelling is novel albeit potentially too specific for the presented setup
- Comprehensive experimental and evaluation setup where the authors curate a dataset of new books (not public due to copyright issues) and compare against multiple reasonable baselines with A/B style human evaluation
- Meaningful insights about the suitability of 3B and 7B models (3B models not suited for the task, while 7B models are)
- The authors say they will release the codebase, which can be helpful, and this work will potentially be used by others in the community

**Reasons To Reject:**

- The fact that it uses perplexity as target for a creative task doesn't sit well with me as lowering entropy or predictability can be considered arguably orthogonal to the intent of creative writing. I understand that it is difficult to come up with a really good measure, but I wouldn't have expected PPL to be the metric of choosing. Using PPL definitely makes sense for a more objective task, where there's less(er) room for creativity, but in this case, I am not sure it "works", as neither the test set is out-of-distribution (doesn't contain articles outside training set genres), and human written stories are almost always exclusively preferred over the proposed model (forcing an inter-variant comparison instead). Maybe you can strengthen the message to be revolve around improving an autonomous model (there are bits and pieces at the moment but it doesn't come off super strong right now).
- Additionally, the exact thresholds in reward function feel quite adhoc to me (I am sure a lot of time was spent on this part) and am concerned that it won't transfer well to another dataset for the same task and someone trying to make it work on their dataset would again be spending a lot of time in figuring these threshold values.
- The above two reasons makes me feel that RL might be shoe-horned for this task and may make a case that an SFT baseline could've worked with enough care, or maybe it just needed a larger dataset or both.

---

> ### Author Response · Authors · 2025-05-28
>
> Thank you for your detailed review! We hope to alleviate some of your concerns and answer your questions.
>
> Addressing concerns:
> R1: Perplexity for creative tasks
> See our response to all reviewers (3).
>
> R1 p2: “[...] not sure it "works", as neither the test set is out-of-distribution (doesn't contain articles outside training set genres)[...]”
> We do not claim that our method generalizes to unseen genres, and rather make efforts to ensure genres are well distributed throughout our train/validation/test sets. Appendix A.1 contains details on genres and data curation methodology. To confirm our method generalizes to producing preferred chapters in unseen books we separate the datasets on the book level and use a maximum of one book for each author. We will clarify this claim in revisions.
>
> R1 part 3: “[...] human written stories are almost always exclusively preferred over the proposed model”
> While true, we don’t believe this is evidence our training methodology doesn’t “work” as it produces chapters preferred over baselines. There are many skills involved in story-writing, many highly impacted by the (untrained) story-generator, thus we don’t expect our method to produce human-level chapters on its own.
>
> Of the long-form story generation work we cite, very few compare directly against human stories and in those that do, human-written stories are preferred. (Yang et al., 2022; 2023; Huot et al., 2025; Xie & Riedl, 2024; Chhun et al., 2022; Peng et al., 2022; Yoo & Cheong, 2024; Chakrabarty et al., 2024). Furthermore, we are the only work amongst these to tackle book-length stories, which we hypothesize to be a harder task for LLMs than short story-writing. We will make this difficulty gap clear in revisions.
>
> R2: Reward Thresholds
> See our response to all reviewers (1).
>
> R3: Improved SFT Baselines
> We selected the hyper-parameters for our SFT baseline based on prior work and a hyper-parameter sweep, evaluated by validation loss. As part of our revisions adding details on how hyper-parameters were selected we can elaborate on this process.
> Per the “larger dataset” comment, one of the core challenges of this setting is the limited and costly nature of large datasets. As such, the standard approach in this domain is prompting-based (lines 47-57), not SFT. It is worth highlighting the difficulty of next-token-prediction for this writing task (lines 20-29), which requires models to balance style and synthesize long-term dependencies. Without a much larger dataset, we hypothesize that SFT is a poor learning objective for this task, motivating our research into alternative training methods.
>
> For fair comparisons, we use the same size dataset for all our models, and find that our method takes better advantage of the limited data. We will clarify this challenge in revisions.
>
> Answering Questions
> Q1: Moving Tables - thank you for the advice, we will move them.
>
> Q2: What other rewards have you tried?
> See our response to all reviewers (1, the last paragraph).
>
> Q3: “Have you tried to use anything other than PPL for your reward?”
> While we considered other reward-model based approaches, prior work has shown little success in robustly applying LLM-based rewards for long-form creative writing. For example, Agent’s Room (~1-2k tokens total) found their LLM-based ranking disagreed with human evaluators who preferred human-written stories (Huot et al., 2024).
>
> Q4: “Table 3 shows mean percent improvement of what exactly? [...]”
> Percent improvement is defined in equation 4 and lines 208-217 - put simply it is the percent improvement in perplexity of the next-chapter, given the generated plan, relative to the baseline perplexity without the plan. The range is theoretically -inf to +inf. Table 3 uses the 7B variant, and Table 14 reports the same metrics with the 3B variant. Future revisions will make this clearer.
>
> Q5: “Why specifically chapter-level? [...] ”
> We generate and evaluate chapters because they are a useful unit for eliciting human preferences, as they don’t require significant time investment (e.g. compared to an entire book) while still containing significant plot/character progression. We are also able to obtain gold-standard high-level chapter summaries that guide the generations without over/underspecification. We have explored generating chunks instead of chapters with some initial success, but obtaining human preferences and guiding story-generations becomes more difficult.
>
> Q6: Model Size vs. Data
> We agree with the hypothesis that data exposure may have a significant effect on model abilities. However, within a model class exposed to the same data (e.g. Qwen 3B & 7B), we provide evidence that increasing model size improves the model’s ability to leverage reasoning. It would be interesting in future work to explore how different base models react to this training paradigm, as recent work (Gandhi et al., 2025) has shown that different model classes have different natural reasoning behaviours.

---

> > ### Comment · Reviewer_oa9L · 2025-06-07
> >
> > Dear authors,
> >
> > Thank you for addressing my queries in detail. I acknowledge reading them and have updated my rating accordingly.

---

### Official Review · Reviewer_jQ4x · 2025-05-13

**Rating:** 6
**Confidence:** 4
**Ethics Flag:** 1

**Summary:**

This paper focuses on the long-form story generation via LLMs, which requires the LLMs to track plot and character arcs consistently and coherently in a long context. Different from previous methods, which rely heavily on the hand-designed prompts, it proposes the next-chapter prediction as the finetuning task and uses the completion likelihood as the reward model for RL training. Pairwise human judgments show that the proposed method is preferred across all metrics.

**Questions To Authors:**

1. Why does Eqn 4 use PPL instead of the log-likelihood?
2. How do you choose the threshold in Eqn 7? How sensitive the RL training is towards these hyperparameters?

**Reasons To Accept:**

1. This proposes the next chapter prediction task for long-form story generation. It uses the story information, such as the global story sketch, previous summary, and character sheet, to generate the next chapter. Instead of supervised fine-tuning, it defines Completion Likelihood Improvement to measure the PPL decrease when given the plan. The reward is used in GRPO to optimize the plan generation model to generate a more informative story plan that can help the generation model.
2. The pairwise human judgements should show that with the story plan, the generated stories are more preferred by humans, and further RL training on the plan generator can make the story better.
3. This paper also provides a story generation dataset, including 30 new books and an average of 130k long contexts. The information, such as a high-level story sketch and story summary, is helpful for long-form story generation.

**Reasons To Reject:**

1. The description of RL for Next Chapter Prediction is confusing. In Section 3, the final output of the Next Chapter Prediction is the generated chapter. However, the objective of RL in Next Chapter Prediction is to optimize the story plan while keeping the chapter generation model πG unchanged. There is a misalignment between the task outcome and the optimization model: it says RL for NCP, but VR-CLI is optimizing the plan instead of directly optimizing the next chapter prediction. It may be better to define the task as the next chapter plan prediction, and the quality of the plan is evaluated by the CLI defined on the next chapter.
2. The lack of a comparison between the supervised finetuning with the RL via VR-CLI in optimizing the plan generation model. To verify the effectiveness of the proposed reward VR-CLI, there should be a supervised fine-tuning baseline. For example, take the chapter summaries from SuperSummary as the plan and fine-tune the story plan model πR.
3. It is difficult to interpret the result in Table 4. The proposed method does not outperform in any metric here. It lacks another qualitative or quantitative analysis on why the story generated by the proposed method is better than the other baselines.

---

> ### Author Response · Authors · 2025-05-28
>
> Thank you for your review! We appreciate you noticed that our approach does not rely on hand-designed prompts. We would like to clarify some of the confusion you mentioned.
>
> Addressing concerns:
> R1: Terminology Confusion
> “The description of RL for Next Chapter Prediction is confusing.” - We define the task as Next-Chapter Prediction because the generated chapters are what we evaluate with human annotators, and compare between model variants. We also do not have gold-standard plans, as the SuperSummary chapter summaries are short and not comparable to the generated reasoning traces (Table 7 shows SuperSummary chapter synopses average 154 tokens, and Table 9 shows reasoning traces are >700 words). Appendix G also shows example reasoning traces that can be qualitatively compared against Table 1’s Next Chapter Synopsis. Similar to prior work that generates outlines or uses agents, we are improving an intermediate representation that is then used to generate the final story continuation (Yang et al., 2022; 2023; Huot et al., 2025). We can make this wording clearer in paper revisions.
>
> R2: Finetuning on Chapter Summaries
> The chapter summaries from SuperSummary (we refer to them as Next Chapter Synopses) are included in the prompt during story-generation, so in effect our “Base” baseline (lines 263-263) is the optimal version of this setup. Our goal is to learn more detailed and expressive plans given this high-level synopsis of the next chapter, and find that including this reasoning improves over this baseline. To illustrate the high-level nature of these synopses, Table 7 shows that these synopses are ~6% of the length of the next chapter. Table 1’s Synopsis and Appendix G can also be compared to qualitatively see the difference in detail. Future work could explore creating this high-level synopsis directly, but as we evaluate the next chapters generated using this synopsis, it is relatively orthogonal to our work.
>
> R3: Interpreting Table 4
> “The proposed method does not outperform in any metric here.” As explained in the description for Table 4 and lines 312-317, the goal of Table 4 is to demonstrate that there are no significant surface-level differences between the generated chapters, across models. We argue in the aforementioned section that this alleviates concerns over biasing human annotators via length or other surface differences. “It lacks another qualitative or quantitative analysis on why the story generated by the proposed method is better than the other baselines.” While Table 4 does not show this, Figure 2, Table 2, Table 15 and lines 299-305 discuss the pairwise human preferences, and show our method is widely preferred over the baselines. How these preferences were collected is presented in Section 7.2.
>
> Answering Questions:
> Q1: “Why does Eqn 4 use PPL instead of the log-likelihood?”
> We use perplexity as the normalization by length makes cross-dataset comparisons and reward thresholds more easily definable.
>
> Q2: How were thresholds chosen, and how sensitive is training?
> See our response to all reviewers, (1).

---

> > ### Comment · Reviewer_jQ4x · 2025-06-05
> >
> > I appreciate the authors' detailed responses, which have addressed most of my concerns. However, I have a few additional points:
> >
> > Regarding the terminology, I find it more intuitive to define the task as optimizing the synopsis, which is referred to as the *"answer"* in this paper. As stated in Lines 203-207, *"a reasoning model generates a reasoning trace t which culminates in a final answer a."* The optimization objective of the reasoning model should focus on the quality of the final answer, rather than the reasoning trace itself. To effectively demonstrate the benefits of fine-tuning the policy model, it is important to show improvements in the output of the policy model (i.e., the detailed plan). I understand the challenge in directly evaluating the quality of the plan, which is why human annotation is performed on the generated chapter. Based on the paper's assumption that a better plan should lead to a better subsequent chapter, this evaluation misalignment can be somewhat mitigated. However, it would be beneficial to frame the problem within a consistent and coherent framework.
> >
> > Regarding Figure 4, if its sole purpose is to show that the generated chapters have similar lengths, I recommend moving it to the appendix. It can serve as a sanity check for human evaluation, ensuring that the evaluation is not influenced by length bias, but it contributes little to demonstrating the advantages of the proposed method and may cause confusion.
> >
> > When using Perplexity (PPL) instead of log-likelihood, there is a risk of gradient explosion due to the exponentiation. This is why PPL is typically used as an evaluation metric rather than a training objective. It is also common to use length-normalized log likelihood in sequence generation tasks.

---

> > > ### Author Response · Authors · 2025-06-10
> > >
> > > Thank you for your feedback! We would like to briefly address your additional points:
> > >
> > > 1) Terminology:
> > >
> > > As mentioned we will make this argument clearer in revisions as the difference may be subtle, but we think it is worth clarifying: Our goal for training our policy model is _not_ to generate plans that humans prefer but rather to generate chapters that humans prefer, via these plans. We have some prior intuition that “good” plans will contain useful information or be intelligible, but this is not a constraint. For example, hypothetically if all annotators hated the plans but the plans produced preferred chapters, they would still meet our objectives. While we  don’t make any claims about  the interpretability of the plans, we view this as an important direction for future work, particularly as a means of explaining model output.
> > >
> > > You mentioned “the paper's assumption that a better plan should lead to a better subsequent chapter” - but in our setting a plan’s “betterness” is defined exclusively by the chapter it induces. Thus a better plan is by definition one that leads to a better chapter. Note that this is separate from the assumption we make with regards to gold-completion likelihood and downstream chapters, defined on lines 199-201.
> > >
> > > Another framing for our work is that of latent-variable modeling: we want to produce a latent variable (plans) that maximize the likelihood of a posterior distribution (chapters). Our policy model is trained to produce a useful latent representation such that the likelihood of the chapters is maximized. We alluded to this framing but can make it and the above clarification clearer in revisions.
> > >
> > > 2) Table 4 - we are happy to move the table to the appendix and add a brief clarification on its purpose.
> > >
> > > 3) Perplexity and risk of gradient explosion:
> > >
> > > While we agree log-likelihood is a valid alternative to perplexity and can mention this in revisions, we don't believe that there is a significant risk of gradient explosion due to 1) the normalizing of this perplexity in our improvement calculation (we also bucket our improvement so there is a min/max value of our reward) and 2) the normalizing of our eventual reward in GRPO's advantage estimation. These drastically shrink the expected variation in our gradient, and note that for our task the perplexity improvements we find are often only a couple percent. As this percent improvement metric is the basis for both defining reward thresholds and tracking training, using an interpretable metric like perplexity is useful.
> > >
> > > To address the concern of training instability for our task we ran a small experiment using 25% of our dataset, for 1/5 the episodes, using the 3B model with two reward settings:
> > >
> > > a) NLL Improvement: max(0, negative log likelihood improvement)
> > >
> > > b) PPL Improvement: max(0, perplexity improvement).
> > >
> > > We approximate the conversion between average NLL Improvement and Average PPL via:
> > >
> > > Average PPL Improvement = $1 - e^{\text{Average Baseline NLL} \cdot \text{Average NLL Improvement}}$
> > >
> > > Where Average Baseline NLL is the average NLL across our validation set (2.551173)
> > >
> > > **Before training:**
> > >
> > > NLL Improvement: 0.0028678
> > >
> > > Pred PPL Improvement: 0.00728956
> > >
> > > Actual PPL Improvement: 0.0074821
> > >
> > > The similarity between these scores validates this conversion process.
> > >
> > > **After training:**
> > >
> > > NLL Improvement: 0.0083329
> > >
> > > Pred PPL Improvement: 0.0210343
> > >
> > > Actual PPL Improvement: 0.02227
> > >
> > > The similarity between these scores indicates that training has resulted in similar gold-chapter likelihood, despite the different improvement metrics.
> > >
> > > Note that training trajectories were not identical and other timesteps would have produced slightly larger differences, but our goal is to show that training is not significantly destabilized. Our hope is that this experiment assuages concerns over this choice of metric for our task; we will add discussion in revisions and will take this into account in future work.

---

### Author Response · Authors · 2025-05-28
**Response to All Reviewers**

As there are some shared comments/questions amongst reviewers we will address them here and reference this section in specific responses to each reviewer.

1) Reward thresholds:  Reviewers asked how our specific reward thresholds were chosen, and about how well these thresholds would generalize to other tasks/datasets.

We selected our reported hyper-parameters from initial experiments investigating the stability of the percent improvements on the validation set, as well as qualitatively assessing the diversity of the generated plans. We find the overall scale of the thresholds (e.g. a 20% percent improvement on our long-generation task would be very large) to be important, but small differences in threshold to have little effect. We would have liked to conduct full experiments investigating the differences in the downstream stories, but conducting human evaluations is an expensive and time-consuming process we reserve for our best performing hyper-parameters. We can elaborate further on this tuning process with examples in revisions of the paper.

We also tried the unbounded version of our reward (Equation 6), but found worse training stability. We hypothesize that amongst sample groups with similar percent-improvement, marginally worse trajectories were unduly discouraged. However, we believe this effect is largely dependent on the percent-improvement-distributions seen throughout training.

We mention on lines 234-236 the justification for these reward thresholds based on initial experiments, but mention in lines 219-224 how different tasks may require different reward formulations (based on our percent-improvement metric). We can include more details on hyper-parameter tuning experiments in paper revisions, adding onto Appendix A.7, but it is difficult for us to recommend any broad policy for defining specific hyper-parameters on someone else’s dataset given the large set of potential use-cases for our reward-formulation.

2) Comparisons with Reward-Model based approaches: Reviewers asked if we attempted other RL approaches (e.g. DPO), or used non-PPL based rewards.

While we considered reward-model based approaches, either by constructing preference datasets (e.g. DPO, RLHF) or training a reward model on annotated data, prior work has shown little success in robustly applying LLM-based rewards for long-form creative writing. For example, in Agent’s Room (~1-2k tokens total) they found their LLM-based ranking disagreed with human evaluators who preferred human-written stories (Huot et al., 2025). As long-context LLM-abilities improve, this avenue of future research may prove more fruitful.

Furthermore, other RL methods (especially those that rely on reward models) are also difficult to apply to our task due to data scarcity and the challenges of long-context creative writing. We briefly describe this distinction with prior tasks on lines 35-46. Another practical consideration is the added training complexity created by using a reward model, which must now be loaded and passed tens of thousands of tokens for each sample.

3) Perplexity as a reward signal: Reviewers asked for our underlying intuition as to why perplexity improvement should correlate with quality

This question revolves around our key assumption (lines 199-201) which states: responses that improve perplexity on the gold dataset will improve downstream generations. We sympathize with the sentiment expressed by some reviewers that perplexity may feel out-of-place for a creative task, but we validate this assumption in two ways: first specifically by showing a significant correlation between percent perplexity improvement and downstream (human) preference (lines 318-324), and secondly more broadly by showing that training with this objective does outperform baselines and produced preferred chapters.

The intuition over VR-CLI’s key assumption’s validity will ultimately depend on the specific use case, but our intuition for using this reward setup for generating plans and next-chapters comes from a collection of prior work showing plans are useful structures for story generation (Yang et al., 2022; 2023; Huot et al., 2025). We hypothesize that a plan that increases the likelihood of the gold next chapter may highlight relevant character details, correctly predict plot events, etc. In contrast, a plan that decreases the likelihood may hallucinate incorrect information or include unlikely plot points. Note that we don’t enforce these constraints, but that they may arise naturally from our reward formulation. We will add this intuition in more detail in paper revisions.

In paper revisions we will also include annotated examples to show how sample plans are useful/distracting for story generation and next-chapter likelihood. This may help readers develop an intuition for why our reward formulation makes sense, and when it would be applicable.

---

### Decision · Program_Chairs · 2025-07-08

**Decision:**

Accept

**Comment:**

The authors uses a clever way to formulate reward function (Verified Rewards via Completion Likelihood Improvement) that allows them to use an unlabeled book dataset as a learning signal for reasoning. They fine-tune a Qwen2.5 model on a next chapter prediction task, where, instead of generating full text, they generate one chapter after the other (with chapters predicted so far being provided as context). The intuition is that model will learn better reasoning and consistency as length increases which is a common pitfall in otherwise long form generation.The authors employ GRPO for training with improvement in per-token perplexity (PPL) as the reward signal. They authors use 30 books for experiment covering four genres: scifi, fantasy, romance, and historical and conduct thorough evaluation. Both quantitative and qualitative results show that the proposed VR-CLI framework leads to better stories compared to the baselines and is favored by humans (although human written stories are still preferred almost exclusively over all).

As several reviewers have noted this is a relatively early and successful adaptation of using RL for open domain challenging creative task while being more effective compared to SFT.  Reviewer oa9L raised several good concerns but authors have addressed them. Overall if the authors address all these in the camera ready and add the clarifications in the extra 1 page, this would make for a strong submission.

[Automatically added comment] At least one review was discounted during the decision process due to quality]